# MISMATCHED NO MORE:
## JOINT MODEL-POLICY OPTIMIZATION FOR MODEL-BASED RL

## ABSTRACT

Many model-based reinforcement learning (RL) methods follow a similar template: fit a model to previously observed data, and then use data from that model for RL or planning. However, models that achieve better training performance (e.g., lower MSE) are not necessarily better for control: an RL agent may seek out the small fraction of states where an accurate model makes mistakes, or it might act in ways that do not expose the errors of an inaccurate model. As noted in prior work, there is an objective mismatch: models are useful if they yield good policies, but they are trained to maximize their accuracy, rather than the performance of the policies that result from them. In this work, we propose a single objective for jointly training the model and the policy, such that updates to either component increase a lower bound on expected return. This joint optimization mends the objective mismatch in prior work. Our objective is a global lower bound on expected return, and this bound becomes tight under certain assumptions. The resulting algorithm (MnM) is conceptually similar to a GAN: a classifier distinguishes between real and fake transitions, the model is updated to produce transitions that look realistic, and the policy is updated to avoid states where the model predictions are unrealistic.

## 1 INTRODUCTION

Much of the appeal of model-based RL is that model learning is a simple and scalable supervised learning problem. Unfortunately, the accuracy of the learned model does not directly correlate with whether the model-based RL algorithm will receive high reward (Farahmand et al., 2017; Lambert et al., 2020). For example, a model might make small mistakes in critical states that cause a policy to take suboptimal actions. Alternatively, a model with large errors may yield a policy that attains high return if the model errors occur in states that the policy never visits.

The underlying problem is that dynamics models are trained differently from how they are used. Typical model-based methods train a model using data sampled from the *real* dynamics (e.g., using maximum likelihood), but apply these models by using data sampled from the *learned* dynamics (Deisenroth & Rasmussen, 2011; Williams et al., 2017; Janner et al., 2019; Hafner et al., 2019). Prior work has identified this *objective mismatch* issue (Farahmand et al., 2017; Luo et al., 2019; Lambert et al., 2020): the model is trained using one objective, but the policy is trained using a different objective. Designing an objective for model training that is guaranteed to improve the expected reward remains an open problem. This paper aims to answer the following question: *How should we train a dynamics model so that it produces high-return policies when used for model-based RL?*

In this paper, we propose a model-based RL algorithm where the model and policy are *jointly optimized* with respect to the same objective. Our objective is a lower bound on the expected return under the true environment dynamics; a slightly more complicated version of this lower bound becomes tight under certain assumptions. Structurally, our algorithm resembles a generative adversarial network (a GAN), in that the model is trained using a discriminator that distinguishes between real and fake transitions. This same discriminator is included in the objective for the policy, and both the model and policy are jointly trained to maximize reward and minimize discriminator accuracy. Thus, the model and policy *cooperate* to produce realistic and high-reward trajectories. Our method stands in contrast to standard model-based RL methods, where it is more common to pit the model *against* the policy (Bagnell et al., 2001; Nilim & El Ghaoui, 2003; Ross & Bagnell, 2012). An consequence of maximizing the lower bound is that the dynamics model does not learn the true dynamics, but rather learns optimistic dynamics that facilitate exploration.

The main contribution of this work is an algorithm, Mismatched no More (MnM), for model-based RL that provably maximizes a lower bound on expected reward. Importantly, this bound becomes tight at optimality under certain assumptions. To the best of our knowledge, this is the first model-based RL objective that is a global lower bound on expected return, and that involves optimizing the model and policy using the same objective. Our algorithm has the unique property of jointly optimizing the policy and model using the same objective. Across a range of tasks, we demonstrate that our method is competitive with prior state-of-the-art methods on benchmark tasks; on certain hard exploration tasks, our method outperforms prior methods based on maximum likelihood estimation.

## 2 RELATED WORK

Model-based RL methods typically fit a dynamics model to observed transitions, and then apply an RL method to that learned model. Most of these methods use maximum likelihood to fit the dynamics model, and then use RL to maximize the expected return under samples from that model (Deisenroth & Rasmussen, 2011; Williams et al., 2017; Chua et al., 2018; Hafner et al., 2019; Janner et al., 2019). The observation that this maximum likelihood objective is not aligned with the RL objective has been noted in prior work (Ziebart, 2010; Talvitie, 2014; Farahmand et al., 2017; Luo et al., 2019; Lambert et al., 2020). This issue is referred to as the *objective mismatch* problem: the model and policy (or planner) are optimized using different objectives. This problem arises in almost all model-based RL approaches, including those that train the model to predict the value function (Oh et al., 2017; Schrittwieser et al., 2020) or that perform planning (Chua et al., 2018; Schrittwieser et al., 2020).

One strategy for mitigating this problem is to modify the model training to improve model accuracy under multi-step rollouts (Joseph et al., 2013; Talvitie, 2014; Venkatraman et al., 2016; Farahmand et al., 2017; Asadi et al., 2018; 2019). A second strategy is penalize the policy for taking transitions where the model is inaccurate (Sorg et al., 2010; Kidambi et al., 2020; Yu et al., 2020b; 2021; Luo et al., 2019). Similar to all these prior methods, our approach will also use a modified reward function to train the policy, but it will also modify how the model is trained such that the model and policy optimize the same objective. A third strategy is to directly optimize the model such that it produces good policies (Okada et al., 2017; Amos et al., 2018; Srinivas et al., 2018; D'Oro et al., 2020; Nikishin et al., 2021), as theoretically analyzed in Grimm et al. (2020). While our aim is the same as these prior methods, our approach will not require differentiating through unrolled model updates or optimization procedures.

Our work builds on prior work that proposes model-based RL objectives that are lower bounds on the true, expected returns. Kearns & Singh (2002) provide a lower bound that holds globally, but is only computable in tabular settings. Luo et al. (2019) provide a lower bound that can be efficiently estimated, but which only holds for nearby policies and models. Our lower bound combines the strengths of these prior works, providing a lower bound that holds globally and can be efficiently estimated in MDPs with continuous states and actions. Unlike any lower bounds from prior work, ours mends the objective mismatch problem.

Our theoretical derivation builds on prior work that casts model-based RL as a two-player game tween a model-player and a policy-player (Bagnell et al., 2001; Nilim & El Ghaoui, 2003; Ross et al., 2011; Rajeswaran et al., 2020). However, whereas prior work pits model and policy compete against one another, our formulation will result in a cooperative game, wherein the model and policy players cooperate in optimizing the *same* objective (a lower bound on the expected return). Our approach, though structurally resembling a GAN, is different from prior work that simply replaces a maximum likelihood model with a GAN model (Bai et al., 2019; Chen et al., 2020; Kurutach et al., 2018).

The most similar prior work is VMBPO (Chow et al., 2020). The mechanics of our method are similar, also learning a dynamics model using a classifier that distinguishes real versus generated rollouts. However, while our method maximizes a lower bound on expected return, VMBPO maximizes a different, risk-seeking objective, which is an *upper* bound on expected return. This different objective can be expressed as the expected return plus the variance of the return, so VMBPO has the undesirable property of preferring policies that receive slight lower return if the variance of the return is much larger (see Appendix A.1). Indeed, most of the components of our method, including classifiers and GAN-like models, have been used in prior work, main contribution of our paper is a precise recipe for combining these components in a way that provably maximizes expected return.

## 3 A UNIFIED OBJECTIVE FOR MODEL-BASED RL

**Notation.** We focus on the Markov decision process with states $s_t$, actions $a_t$, initial state distribution $p_0(s_0)$, positive reward function $r(s_t, a_t) > 0$, and dynamics $p(s_{t+1} \mid s_t, a_t)$. Our aim is to learn a control policy $\pi_\theta(a_t \mid s_t)$ with parameters $\theta$ that maximizes the expected discounted return:

$$\max_\theta \mathbb{E}_{\pi_\theta} \left[ \sum_{t=0}^\infty \gamma^t r(s_t, a_t) \right]. \tag{1}$$

We use transitions $(s_t, a_t, r_t, s_{t+1})$ collected from the (real) environment to train the dynamics model $q_\theta(s_{t+1} \mid s_t, a_t)$, and use transitions sampled from this learned model to train the policy. To simplify notation, we will define a trajectory $\tau$ to be the sequence of states and actions visited in an episode: $\tau \triangleq (s_0, a_0, s_1, a_1, \cdots)$. We then define $R(\tau) \triangleq \sum_{t=0}^\infty \gamma^t r(s_t, a_t)$ as the discounted return of a trajectory. Finally, we define two distributions over trajectories. First, $p^\pi(\tau)$ is the distribution over trajectories when policy $\pi_\theta$ interacts with dynamics $p(s' \mid s, a)$; $q^\pi(\tau)$ is the distribution over trajectories when policy $\pi_\theta$ interacts with the learned dynamics $q_\theta(s_{t+1} \mid s_t, a_t)$:

$$p^\pi(\tau) = p_0(s_0) \prod_{t=0}^\infty p(s_{t+1} \mid s_t, a_t) \pi_\theta(a_t \mid s_t), \quad q^\pi(\tau) = p_0(s_0) \prod_{t=0}^\infty q_\theta(s_{t+1} \mid s_t, a_t) \pi_\theta(a_t \mid s_t).$$

**Desiderata.** Our aim is to design an objective $\mathcal{L}(\theta)$ that can be used to *jointly* optimize both the policy ($\pi_\theta(a_t \mid s_t)$) and the dynamics model ($q(s_{t+1} \mid s_t, a_t)$), and which is a lower bound on the expected return in the true environment.

**An objective for model-based RL.** We now introduce an objective that achieves these aims. Our objective will be the policy's reward when interacting with the learned model, but using a different reward function. The new reward function augments the task reward with an additional term that measures the difference between the learned model and the real environment. We define our objective

$$\mathcal{L}(\theta) \triangleq \mathbb{E}_{q^{\pi_\theta}(\tau)} \left[ \sum_{t=0}^\infty \gamma^t \tilde{r}(s_t, a_t, s_{t+1}) \right], \tag{2}$$

where the modified reward function is defined as

$$\tilde{r}(s_t, a_t, s_{t+1}) \triangleq (1 - \gamma) \log r(s_t, a_t) + \log \left( \frac{p(s_{t+1} \mid s_t, a_t)}{q(s_{t+1} \mid s_t, a_t)} \right) - (1 - \gamma) \log(1 - \gamma). \tag{3}$$

This objective maximizes an *augmented* reward under the *learned* dynamics. The augmented reward function $\tilde{r}$ penalizes the policy for taking transitions that are unlikely under the true dynamics model, similar to prior work (Eysenbach et al., 2020a; Yu et al., 2021). Later, we will show that we can estimate this augmented reward *without knowing the true environment dynamics* by using a GAN-like classifier. We will optimize this lower bound with respect to both the policy $\pi_\theta(a_t \mid s_t)$ and the dynamics model $q_\theta(s_{t+1} \mid s_t, a_t)$. For the policy, this optimization entails performing RL to maximize the modified reward using samples from the learned model; the only difference from prior work is the modification to the reward function. Training the dynamics model using this objective is very different from standard maximum likelihood training, and instead resembles a GAN. The model is optimized to sample trajectories that both have high reward (i.e., $\log r$ is large) and are similar to real dynamics (i.e., $\log \frac{p}{q}$ is large). This objective differs from VMBPO (Chow et al., 2020) by taking the $\log(\cdot)$ of the original reward functions; our experiments demonstrate that excluding this component invalidates our lower bound and results in learning suboptimal policies.

Our objective has two properties that make it particularly useful. First, the model and the policy are trained using exactly the same objective: updating the model not only increases the objective for the model, but also increases the objective for the policy. Note that this is very different from prior work, where training the model to be more accurate (increase likelihood) can decrease the policy's expected return under that model. Second, our objective is a lower bound on the expected return. This property gives us a guarantee on how well the learned policy will perform when deployed on the real environment. To state this result formally, we will take the logarithm of the expected return. Of course, maximizing the $\log(\cdot)$ of the expected return is equivalent to maximizing the expected return.

**Theorem 3.1.** *The following bound holds for **any** dynamics $q(s_{t+1} \mid s_t, a_t)$ and policy $\pi(a_t \mid s_t)$:*

$$\log \mathbb{E}_\pi \left[ \sum_{t=0}^\infty \gamma^t r(s_t, a_t) \right] \geq \mathcal{L}(\theta).$$

The proof is presented in Appendix A.3. Note that the expected return under the learned model, which most prior model-based RL methods use to train the policy, is not a lower bound on the expected return. To the best of our knowledge, this is the first global (unlike Luo et al. (2019)) and efficiently-computable (unlike Kearns & Singh (2002)) lower bound for model-based RL.

Sec. 4 will introduce an algorithm to maximize this lower bound. While this lower bound may not be tight, experiments in Sec. 5 demonstrate that optimizing this first lower bound yields policies that achieve high reward across a wide range of tasks.

**Tightening the lower bound.** We now introduce a modification to our lower bound that does make the bound tight. This new lower bound will be more complex than the one introduced above and we have not yet successfully designed an algorithm for maximizing it. Nonetheless, we believe that presenting the bound may prove useful for the design of future model-based RL algorithms.

We will use $\mathcal{L}_\gamma(\theta)$ to denote this new lower bound. In addition to the policy and dynamics, this bound will also depend on a time-varying discount, $\gamma_\theta(t)$, in place of the typical $\gamma^t$ term. Similar learned discount factors have been studied in previous work on model-free RL (Rudner et al., 2021). We define this objective as follows:

$$\mathcal{L}_\gamma(\theta) \triangleq \mathbb{E}_{q^{\pi_\theta}(\tau)}\left[\sum_{t=0}^{\infty} \gamma_\theta(t)\tilde{r}_\gamma(s_t, a_t, s_{t+1})\right], \qquad (4)$$

where the augmented reward is now defined as

$$\tilde{r}_\gamma(s_t, a_t, s_{t+1}) \triangleq \log r(s_t, a_t) + \frac{1 - \Gamma_\theta(t-1)}{\gamma_\theta(t)} \log\left(\frac{p(s_{t+1} \mid s_t, a_t, s_{t-1}, a_{t-1}, \cdots)}{q_\theta(s_{t+1} \mid s_t, a_t, s_{t-1}, a_{t-1}, \cdots)}\right) + \log\left(\frac{\gamma^t}{\gamma_\theta(t)}\right),$$

and $\Gamma_\theta(t) = \sum_{t'=0}^{t} \gamma_\theta(t')$ is the CDF of the learned discount function (i.e., $\gamma_\theta(t)$ is a probability distribution over $t$.). This new lower bound, which differs from our main lower bound by the learnable discount factor, does provide a tight bound on the expected return objective:

**Lemma 3.2.** *Let an arbitrary policy $\pi(a_t \mid s_t)$ be given. The objective $\mathcal{L}_\gamma(\theta)$ is also a lower bound on the expected return objective, $\log \mathbb{E}_\pi\left[\sum_{t=0}^{\infty} \gamma^t r(s_t, a_t)\right] \geq \mathcal{L}_\gamma(\theta)$, and this bound becomes tight at optimality:*

$$\log \mathbb{E}_\pi\left[\sum_{t=0}^{\infty} \gamma^t r(s_t, a_t)\right] = \max_{q^\pi(\tau), \gamma_\theta(t)} \mathcal{L}_\gamma(\theta).$$

The proof is presented in Appendix A.4. One important limitation of this result is that the learned dynamics that maximize this lower bound to make the bound tight may be non-Markovian. Intriguingly, this analysis suggests that using non-Markovian models, such as RNNs and transformers, may accelerate learning on Markovian tasks. This paper does not propose an algorithm for optimizing this more complex lower bound.

**The optimal dynamics are optimistic.** We now return to analyzing the simpler lower bound ($\mathcal{L}(\theta)$ in Eq. 2). In stochastic environments, the dynamics model that optimizes this lower bound is not equal to the true environment dynamics. Rather, it is biased towards sampling trajectories with high return. Ignoring parametrization constraints, the dynamics model that optimizes our lower bound is $q^*(\tau) = \frac{p(\tau)R(\tau)}{\int p(\tau')R(\tau')d\tau'}$ (proof in Appendix A.4.). We hypothesize that the optimism in the dynamics model will accelerate policy optimization, a hypothesis we test in Sec. 5.1.

Would the optimistic dynamics overestimate the policy's return, violating Lemma 3.1? This is not quite what our method does. Rather, our method estimates the *augmented* reward using the optimistic dynamics model, and the reward augmentation compensates for the optimism in the dynamics model.

## 4 MISMATCHED NO MORE

The previous section presented a single (global) lower bound ($\mathcal{L}$ from Eq. 2) for jointly optimizing the policy and the dynamics model. In this section, we develop a practical algorithm for optimizing this lower bound. We call our method MISMATCHED NO MORE (MnM) because the policy and model optimize the same objective, thereby resolving the objective mismatch problem noted in prior work. The main challenge in optimizing this bound is that the augmented reward function depends on

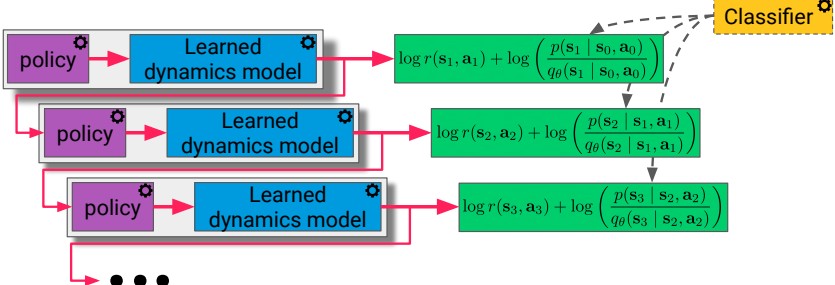

Figure 1: **Mismatched No More** is a model-based RL algorithm that learns a policy, dynamics model, and classifier. The classifier distinguishes real transitions from model transitions. The policy and dynamics model are jointly optimized to sample transitions that yield high return and look realistic, as estimated by the classifier.

the transition probabilities of the real environment, $p(s_{t+1} \mid s_t, a_t)$, which are unknown. We address this challenge by learning a classifier (Sec. 4.1), and then describe the precise update rules for the policy, dynamics model, and classifier (Sec. 4.2).

## 4.1 ESTIMATING THE AUGMENTED REWARD FUNCTION

To estimate the augmented reward function, which depends on the transition probabilities of the real environment, we learn a classifier that distinguishes real transitions from fake transitions. This approach is similar to GANs (Goodfellow et al., 2014) and similar to prior work in RL (Eysenbach et al., 2020a; Yu et al., 2021). We use $C_\phi(s_t, a_t, s_{t+1}) \in [0, 1]$ to denote the classifier. For an optimal classifier, we can use the classifier's predictions to estimate the augmented reward function:

$$\tilde{r}(s_t, a_t, s_{t+1}) = \log r(s_t, a_t) + \underbrace{\log \frac{C_\phi(s_t, a_t, s_{t+1})}{1 - C_\phi(s_t, a_t, s_{t+1})}}_{\approx \log p(s_{t+1}|s_t,a_t) - \log q_\theta(s_{t+1}|s_t,a_t)} . \tag{5}$$

The approximation above reflects function approximation error in learning the classifier.

We now present our complete method, which trains three components: a classifier, a policy, and a dynamics model. Our method alternates between *(1)* updating the policy (by performing RL using model experience with augmented rewards) and *(2)* updating the dynamics model and classifier (using a GAN-like objective). In describing the loss functions below, we use the superscripts $(\cdot)^{\text{real}}$ and $(\cdot)^{\text{model}}$ to denote transitions that have been sampled from the true environment dynamics or the learned dynamics function. To reduce clutter, we omit the superscripts when unambiguous.

## 4.2 UPDATING THE MODEL, POLICY, AND CLASSIFIER

**Updating the policy.** The policy is optimized to maximize the augmented reward on transitions sampled from the learned dynamics model. While this optimization can be done using any RL algorithm, including on-policy methods, we will focus on an off-policy actor-critic method.

We define the Q function as sum of *augmented* rewards under the learned dynamics model:

$$Q(s_t, a_t) \triangleq \mathbb{E}_{\substack{\pi(a_t|s_t), \\ q_\theta(s_{t+1}|s_t,a_t)}} \left[ \sum_{t'=t}^{\infty} \gamma^{t'-t} \tilde{r}(s_{t'}, a_{t'}) \mid s_t = s_t, a_t = a_t \right]. \tag{6}$$

We approximate the Q function using a neural network $Q_\psi(s_t, a_t)$ with parameters $\phi$. We train the Q function using the TD loss on transitions sampled from the *learned* dynamics model:

$$\mathcal{L}_Q(s_t, a_t, r_t, s_{t+1}^{\text{model}}; \psi) = \left( Q_\psi(s_t, a_t) - \lfloor y_t \rfloor_{\text{sg}} \right)^2, \tag{7}$$

where $\lfloor \cdot \rfloor_{\text{sg}}$ is the stop-gradient operator and $y_t = \tilde{r}(s_t, a_t, s_{t+1}^{\text{model}}) + \gamma \mathbb{E}_{\pi(a_{t+1}|s_{t+1}^{\text{model}})} \left[ Q_\psi(s_{t+1}^{\text{model}}, a_{t+1}) \right]$. The augmented reward, $\tilde{r}$, is estimated using the learned classifier. To estimate the corresponding value function, we use a 1-sample approximation: $V_\psi(s_t) = Q_\psi(s_t, a_t \sim \pi_\theta(a_t \mid s_t))$. The policy is trained to maximize the expected future (augmented) return, as estimated by the Q function:

$$\max_\theta \mathcal{L}_\pi(s_t; \theta) \triangleq \mathbb{E}_{\pi_\theta(a_t|s_t)} \left[ \tilde{Q}_\psi(s_t, a_t) \right]. \tag{8}$$

In our implementation, we regularize the policy by adding an additional entropy regularizer. Following prior work (Fujimoto et al., 2018), we maintain two Q functions and two target Q functions, use the minimum of the two target Q functions to compute the TD target. See Appendix C for details.

---

**Algorithm 1 Mismatched no More (MnM)** is an algorithm for model-based RL. The method alternates between training the policy on experience from the learned dynamics model with augmented rewards and updating the model+classifier using a GAN-like loss. Updates are gradient steps with the Adam optimizer.

---

1: **while** not converged **do**
2:     Sample experience from learned model and modify rewards using the classifier (Eq. 5).
3:     Update policy and Q function using the model experience and augmented rewards (Eq.s 8 and 7).
4:     Update model and classifier using GAN-like losses (Eq.s 9 and 10).
5:     (Infrequently) Sample experience from real model.
6: **return** policy $\pi_\theta(a_t \mid s_t)$.

---

**Updating the classifier.**     We train the classifier to distinguish real versus model transitions using the standard cross entropy loss:

$$\max_\phi \mathcal{L}_C(s_t^{\text{real}}, a_t^{\text{real}}, s_{t+1}^{\text{real}}, s_{t+1}^{\text{model}}; \phi) \triangleq \log C_\phi(s_t^{\text{real}}, a_t^{\text{real}}, s_{t+1}^{\text{real}}) + \log \left(1 - C_\phi(s_t^{\text{real}}, a_t^{\text{real}}, s_{t+1}^{\text{model}})\right). \quad (9)$$

Note that the real transition $(s_t^{\text{real}}, a_t^{\text{real}}, s_{t+1}^{\text{real}})$ and model transition $(s_t^{\text{real}}, a_t^{\text{real}}, s_{t+1}^{\text{model}})$ have the same initial state and initial action.

**Updating the dynamics model.**     To optimize the dynamics model, we rewrite the lower bound in terms of a single transition (derivation in Appendix A.6):

$$\mathcal{L}_q(s_t^{\text{real}}, a_t^{\text{real}}; \theta) = \mathbb{E}_{s_{t+1}^{\text{model}} \sim q_\theta(s_{t+1}|s_t^{\text{real}}, a_t^{\text{real}})} \left[ V_\psi(s_{t+1}^{\text{model}}) + \log \frac{C_\phi(s_t^{\text{real}}, a_t^{\text{real}}, s_{t+1}^{\text{model}})}{1 - C_\phi(s_t^{\text{real}}, a_t^{\text{real}}, s_{t+1}^{\text{model}})} \right]. \quad (10)$$

The approximation above reflects approximation error in learning the optimal classifier. This approximation is standard in prior work on GANs (Goodfellow et al., 2014) and adversarial inference (Donahue et al., 2016; Dumoulin et al., 2016). The procedure for optimizing the dynamics model and the classifier resembles a GAN (Goodfellow et al., 2014): the classifier is optimized to distinguish real transitions from model transitions, and the model is updated to fool the classifier (and increase rewards). However, *our method is not equivalent to simply replacing a maximum likelihood model with a GAN model*. Indeed, such an approach would not optimize a lower bound on expected return. Rather, our model objective includes an additional value term and our policy objective includes an additional classifier term. These changes enable the model and policy to optimize the same objective, which is a lower bound on expected return.

**Algorithm summary.**     We summarize the method in Alg. 1 and provide an illustration in Fig. 1. Implementing MnM on top of a standard model-based RL algorithm is straightforward. First, create an additional classifier network. Second, instead of using the maximum likelihood objective to train the model, use the GAN-like objective in Eq. 10 to update both the model and the classifier. Third, add the classifier's logits to the predicted rewards (Eq. 5). Following prior work (Janner et al., 2019), we learn a neural network to predict the true environment rewards.

## 5  EXPERIMENTS

We present two sets of experiments. Our first set of experiments studies the importance of different components of MnM . Second, we study how MnM compares with prior model-based RL algorithms on challenging, robotic control tasks. To compare policies learned by different algorithms, we will evaluate the policies using the true environment dynamics, not the learned dynamics model.

### 5.1  UNDERSTANDING THE LOWER BOUND AND THE LEARNED DYNAMICS

In this section, we begin by studying the seemingly contradictory attributes of our method: optimistic dynamics and pessimistic policies, and end by confirming that together these components optimize an increasingly tight lower bound on the expected return.

Our theory suggests that MnM should work best in settings with stochastic dynamics and challenging exploration requirements, as the dynamics model should tilt the true dynamics to make the stochasticity more favorable for solving the task. We use a 10x10 gridworld with highly stochastic dynamics and a sparse reward function. The results, shown in Fig. 2a *(Left)*, show that MnM outperforms both Q-learning and VMBPO. In line with our theory, the dynamics learned by MnM (Fig. 2a *(Right)*) alter

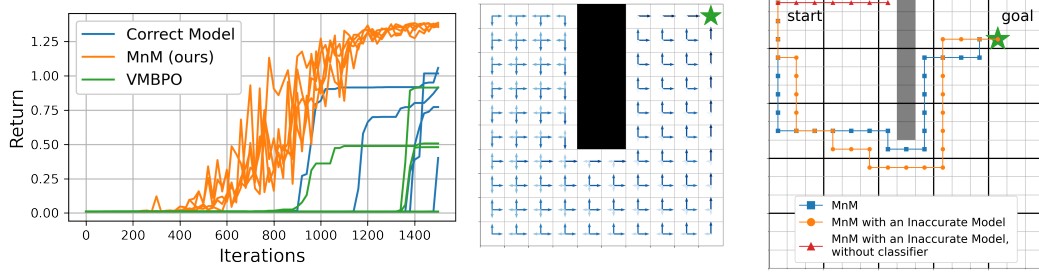

(a) Stochastic Gridworld            (b) Inaccurate Models

Figure 2: **Two Didactic Experiments.** *(Left)* We apply MnM to a navigation task with transition noise that moves the agent to neighboring states with equal probability. MnM solves this task more quickly than Q-learning and VMBPO. The dynamics learned by MnM are different from the real dynamics, changing the transition noise (blue arrows) to point towards the goal. *(Right)* We simulate function approximation by a learning model that is forced to make the same predictions for groups of $3 \times 3$ states, resulting in a model that is inaccurate around obstacles. The classifier term compensates for this function approximation error by penalizing the policy for navigating near obstacles.

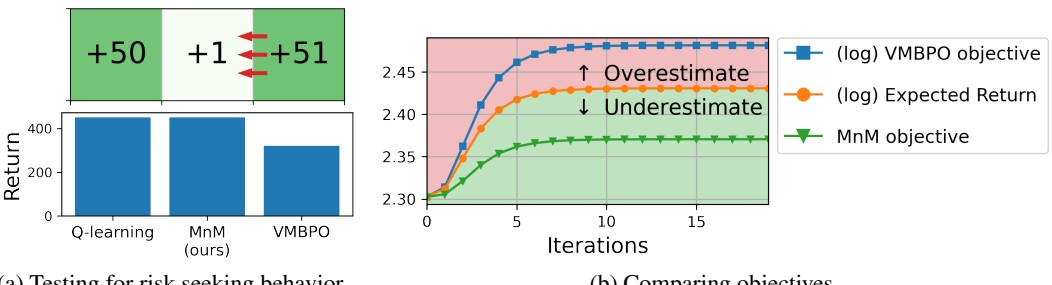

(a) Testing for risk seeking behavior.          (b) Comparing objectives.

Figure 3: **Lower Bounds and Risk Seeking.** *(Left)* On a simple 3-state MDP with stochastic transition in one state (red arrows), MnM converges to the reward-maximizing policy while VMBPO learns a strategy with lower rewards and higher variance (as predicted by theory). *(Right)* We apply value iteration to the gridworld from Fig. 2a to analytically compute various objectives. As predicted by our theory, the MnM objective is a lower bound on the expected return, whereas the VMBPO objective overestimates the expected return.

the environment stochasticity to lead the agent towards the goal, increasing the probability of collecting high-reward experience. Of course, we use the true environment dynamics, not the optimistic dynamics model, for evaluating the policies. While VMBPO also learns optimistic dynamics, it omits log-transformation of the reward function (which encourages pessimistic behavior), a difference that has a large effect on this task.

Our augmented reward function contains two crucial components, *(1)* the classifier term and *(2)* the logarithmic transformation of the reward function. We test the importance of the classifier term in correcting for inaccurate models. To do this, we limit the capacity of the MnM dynamics model so that it makes "low-resolution" predictions, forcing all states in $3 \times 3$ blocks to have the same dynamics. We will use the gridworld shown in Fig. 2b, which contains obstacles that occur at a finer resolution than the model can detect. When the "low resolution" dynamics model makes predictions for states near the obstacle, it will average together some states with obstacles and some states without obstacles. Thus, the model will (incorrectly) predict that the agent always has some probability of moving in each direction, even if that direction is actually blocked by an obstacle. However, the classifier (whose capacity we have not limited) detects that the dynamics model is inaccurate in these states and so the augmented reward is much lower at these states. Thus, MnM is able to solve this task despite the inaccurate model; an ablation of MnM that removes the classifier term attempts to navigate through the wall and fails to reach the goal.

We then test the importance of the logarithmic transformation by comparing MnM to VMBPO, which includes the classifier term but omits the logarithmic transformation. We hypothesize that VMBPO's deviation from our lower bound will cause it to exhibit risk seeking behavior. To test this hypothesis, we use the 3-state MDP in Fig. 3a *(top)* where numbers indicate the reward at each state. While moving to the right state yields slightly higher rewards, "wind" knocks the agent out of this state with probability 50% so the reward-maximizing strategy is to move to the left state. While MnM

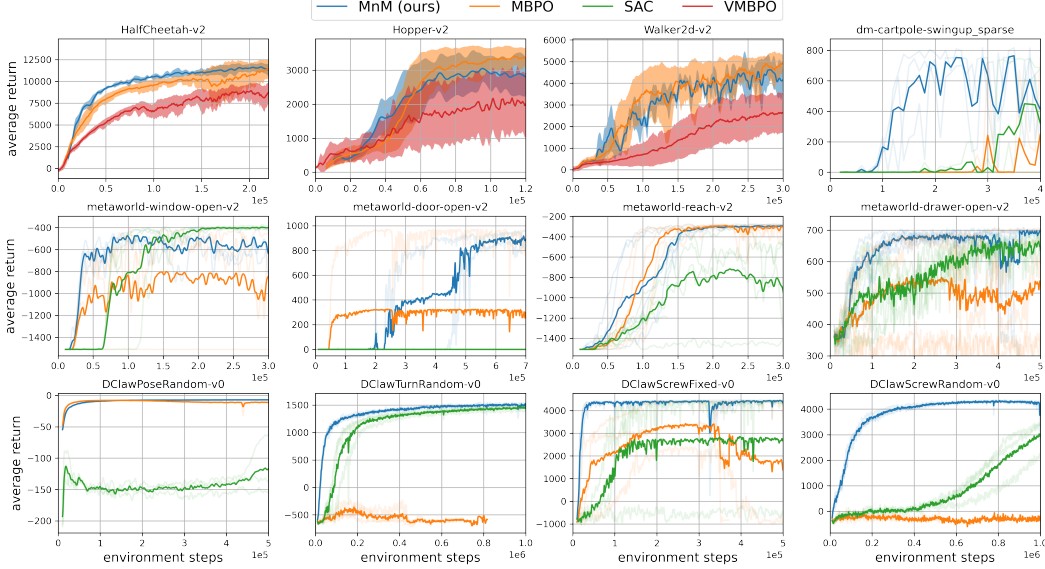

Figure 5: **Comparison on Robotics Tasks**: We compare MnM to MBPO and SAC on simulated control tasks. On the benchmark locomotion tasks *(top left)*, MnM performs comparably with MBPO. On many of the other tasks with sparse rewards that pose an exploration challenge, MnM outperforms both MBPO and the model-free baseline. These experiments suggest that maximizing a well defined bound on expected return, as done by our method, can lead to improved performance on difficult tasks.

MnM learns the reward-maximizing strategy, VMBPO learns a policy that goes to the right state and receives lower returns (with much higher variance).

Finally, we study how the MnM objective compares to alternative objectives. We use the gridworld from Fig. 2a and use a version of MnM based on value iteration to avoid approximation error. Plotting the MnM objective in Fig. 3b, we observe that it is always a lower bound on the (log) expected return, as predicted by our theory. Ablations of MnM to omit the reward augmentation or even just the log transformation (i.e., VMBPO) overestimate the expected return.

## 5.2 COMPARISONS ON ROBOTICS TASKS

Our next experiments use continuous-control robotic tasks to answer two questions. We first investigate whether MnM performs at least comparably with prior work. We then study tasks with sparse rewards and more challenging exploration, where we suspect the optimistic dynamics learned by MnM may be beneficial. We illustrate a subset of the environments in Fig. 4 and include implementation details in Appendix C.2. One detail of note is that we omit the reward augmentation (Eq. 3) for MnM during these experiments, as it hinders exploration leading to lower returns. We use MBPO (Janner et al., 2019) as a baseline for model-based RL because it achieves state-of-the-art results and is a prototypical example of model-based RL algorithms that use maximum likelihood models.

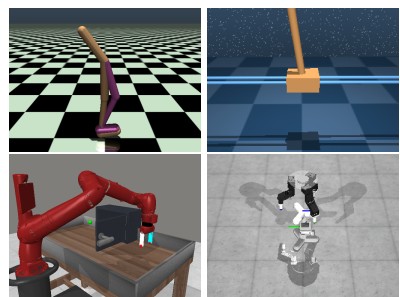

Figure 4: **Environments**: Our experiments included tasks from four benchmarks: (clockwise from top-left) OpenAI Gym, DM Control, Metaworld, and ROBEL.

Our first comparison uses three locomotion tasks from the OpenAI Gym benchmark (Brockman et al., 2016), which has become the standard benchmark for model-based RL algorithms. Thes tasks have dense rewards and pose no significant exploration challenge, so we do not expect MnM to outperform prior methods. The results (Fig. 5 *(top left)*) show that MnM performs roughly on par with MBPO.

Tasks with sparse rewards, complex contact dynamics, and those requiring hard exploration often present a challenge for model-based RL algorithms, which are liable to exploit inaccuracies in the learned dynamics model. Our next experiment evaluates MnM on control tasks used in prior work that demonstrate these properties. These tasks the sparse-reward cartpole task from the DM Control benchmark (Tassa et al., 2018), four manipulation tasks from the Metaworld benchmark (Yu et al., 2020a), and four dexterous manipulation tasks from the ROBEL benchmark (Ahn et al., 2020). See

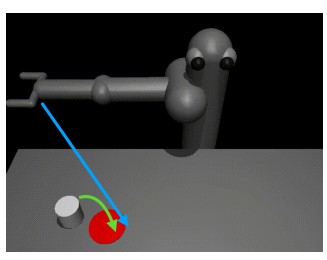
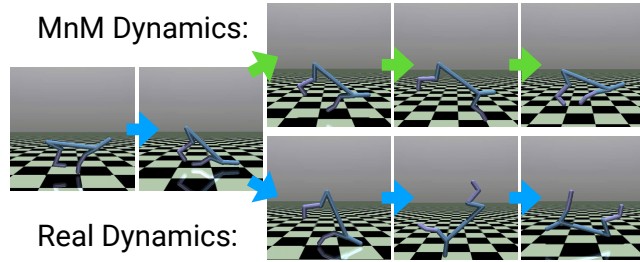

Figure 7: **Optimistic Dynamics**: *(Left)* On the `Pusher-v2` task, the MnM dynamics model makes the puck move towards the puck move towards the gripper before being grasped. *(Right)* On the `HalfCheetah-v2` task, the MnM dynamics model helps the agent stay upright after tripping.

Appendix C for experiment details. The results shown in Fig. 5 indicate that the complex environment dynamics of these tasks can cause prior model-based algorithms (MBPO) to perform worse than model-free algorithms (SAC), both in terms of asymptotic performance and sample complexity. Nonetheless, we observe that MnM frequently outperforms all prior methods and, more importantly, it *consistently* does well across all tasks. We include ablation experiments, including a comparison to VAML (Farahmand et al., 2017), in Appendix B.

To investigate the benefits of MnM over prior model-based methods, we logged the Q values throughout training and visualized them for the `metaworld-drawer-open-v2` task in Fig. 6. For fair comparison, we use Q values corresponding to just the task reward, omitting the logarithmic transformation and classifier term typically used by MnM. Fig. 6 shows that MnM yields Q values that are more accurate and more stable than MBPO. This figure suggests that MBPO may be exploiting inaccuracies in the learned model.

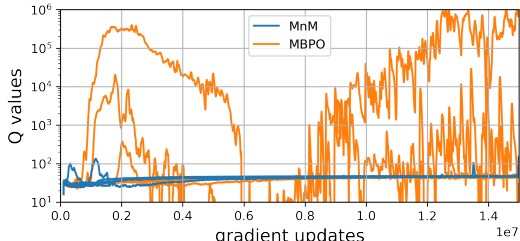

Figure 6: **Model exploitation**: The very large Q values of MBPO suggest model exploitation, which our method appears to avoid.

Finally, we visualize the dynamics learned by MnM on two robotic control tasks. These tasks have deterministic dynamics, so our theory would predict that an idealized version of MnM would learn a dynamics model exactly equal to the deterministic dynamics. However, our implementation relies on function approximation (neural networks) to learn the dynamics, and the limited capacity of function approximators makes otherwise-deterministic dynamics appear stochastic. On the `Pusher-v2` task, the MnM dynamics cause the puck to move towards the robot arm even before the arm has come in contact with the puck. While this movement is not physically realistic, it may make the exploration problem easier. On the `HalfChetah-v2` task, the MnM dynamics increase the probability that the agent remains upright after tripping, likely making it easier for the agent to learn how to run. We expect that the implicit stochasticity caused by function approximation to be especially important for real-world tasks, where the complexity of the real dynamics often dwarfs the capacity of the learned dynamics model.

## 6 CONCLUSION

The main contribution of this paper is an approach to model-based RL where the policy and dynamics model are jointly optimized using the same objective. Unlike prior work, our objective is a global lower bound on the standard expected return objective. Our approach not only tells users how to train their dynamics model, but also guarantees to them that updating their model (using our objective) will result in a better policy. We therefore believe that this *joint optimization* will ease and accelerate the design of future model-based RL algorithms. We suspect that the tools presented in this paper may prove useful for solving tasks that require extensive exploration or long-horizon planning.

**Limitations.** Compared with methods based on maximum likelihood, our method requires learning an additional classifier and balancing the capacity of that classifier against the capacity of the dynamics model. Unlike VMBPO, our method does not have a single hyperparameter to control the gap between the proposed objective and the expected return objective.

**Ethics Statement.** This paper works to advance model based reinforcement learning, which can enable data driven policies to learn from smaller datasets than their model-free counterparts. In doing so, this expands the range of applications for which reinforcement learning can be used. Many of these low data regimes, which can include things such as health care, require not just data efficiency but also safe online fine-tuning. It is important to note that although our method optimizes a lower-bound on the expected returns in theory, this lower-bound utilizes function approximators, which do not come with guarantees in practice.

**Reproducibility.** Our appendix contains proofs for all theoretical claims, the hyperparameters and implementation details of each algorithm used, ablation experiments illustrating the expected behavior of our method under varying circumstances of interest, implementation details that we did and did not find helpful, and finally the necessary details to recreate each environment used in the paper. Furthermore, we will be publicly releasing the code for our didactic experiments.

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

# A  PROOFS AND ADDITIONAL ANALYSIS

## A.1  VMBPO MAXIMIZES AN UPPER BOUND ON RETURN

While MnM aims to maximize the (log) of the expected return, VMBPO aims to maximize the expected *exponentiated* return:

$$\text{MnM:}\quad \log \mathbb{E}_\pi \left[ \sum_{t=0}^{\infty} \gamma^t r(s_t, a_t) \right], \qquad \text{VMBPO:}\quad \log \mathbb{E}_\pi \left[ e^{\eta \sum_{t=0}^{\infty} \gamma^t r(s_t, a_t)} \right],$$

where $\eta > 0$ is a temperature term used by VMBPO. Note that maximizing the log of the expected return, as done by MnM, is equivalent to maximizing the expected return, as the function $\log(\cdot)$ is monotone increasing. However, maximizing the log of the expected *exponentiated* return, as done by VMBPO, is not equivalent to maximizing the expected return. Rather, it corresponds to maximizing a sum of the expected return and the *variance* of the return (Mihatsch & Neuneier, 2002, Page 272):

$$\frac{1}{\eta} \log \mathbb{E}_\pi \left[ e^{\eta \sum_{t=0}^{\infty} \gamma^t r(s_t, a_t)} \right] = \mathbb{E}_\pi \left[ \sum_{t=0}^{\infty} \gamma^t r(s_t, a_t) \right] + \frac{\eta}{2} \text{Var}_\pi \left[ \sum_{t=0}^{\infty} \gamma^t r(s_t, a_t) \right] + \mathcal{O}(\eta^2).$$

Thus, in environments with stochastic dynamics or rewards (e.g., the didactic example in Fig. 3), VMBPO will prefer to receive lower returns if the variance of the returns is much higher. We note that the expected *exponentiated* return is an *upper* bound on the expected return:

$$\log \mathbb{E}_\pi \left[ e^{\eta \sum_{t=0}^{\infty} \gamma^t r(s_t, a_t)} \right] \geq \eta \mathbb{E}_\pi \left[ \sum_{t=0}^{\infty} \gamma^t r(s_t, a_t) \right].$$

This statement is a direct application of Jensen's inequality. The bound holds with a strict inequality in almost all MDPs. The one exception is trivial MDPs where all trajectories have exactly the same return. Of course, even a random policy is optimal for these trivial MDPs.

## A.2  HELPER LEMMAS

We start by introducing a simple identity that will help handle discount factors in our analysis.

**Lemma A.1.** *Define $p(H) = \textsc{Geom}(1 - \gamma)$ as the geometric distribution. Let discount factor $\gamma \in (0, 1)$ and random variable $x_t$ be given. Then the following identity holds:*

$$\mathbb{E}_{p(H)} \left[ \sum_{t=0}^{H} x_t \right] = \sum_{t=0}^{\infty} \gamma^t x_t.$$

The proof involves substituting the definition of the Geometric distribution and then rearranging terms.

*Proof.*

$$\begin{aligned}
\mathbb{E}_{p(H)} \left[ \sum_{t=0}^{H} x_t \right] &= (1 - \gamma) \sum_{H=0}^{\infty} \gamma^H \sum_{t=0}^{H} x_t \\
&= (1 - \gamma) \left( x_0 + \gamma(x_0 + x_1) + \gamma^2(x_0 + x_1 + x_2) + \cdots \right) \\
&= (1 - \gamma) \left( x_0(1 + \gamma + \gamma^2 + \cdots) + x_1(\gamma + \gamma^2 + \cdots) + \cdots \right) \\
&= (1 - \gamma) \left( x_0 \frac{1}{1 - \gamma} + x_1 \frac{\gamma}{1 - \gamma} + x_2 \frac{\gamma^2}{1 - \gamma} + \cdots \right) \\
&= \sum_{t=0}^{\infty} \gamma^t x_t.
\end{aligned}$$

$\square$

The second helper lemma describes how the discounted expected return objective can be written as the expected *terminal* reward of a mixture of finite-length episodes.

**Lemma A.2.** *Define $p(H) = \text{GEOM}(1 - \gamma)$ as the geometric distribution, and $p(\tau \mid H)$ as a distribution over length-$H$ episodes. We can then write the expected discounted return objective as follows:*

$$\mathbb{E}_{p(\tau|H=\infty)} \left[ \sum_{t=0}^{\infty} \gamma^t r(s_t, a_t) \right] = \frac{1}{1 - \gamma} \mathbb{E}_{p(H)} \left[ \mathbb{E}_{p(\tau|H=H)} \left[ r(s_H, a_H) \right] \right] \tag{11}$$

$$= \frac{1}{1 - \gamma} \iint p(H) p(\tau \mid H = H) r(s_H, a_H) d\tau dH. \tag{12}$$

*Proof.* The first identity follows from the definition of the geometric distribution. The second identity writes the expectations as integrals, which will make future analysis clearer. $\square$

### A.3 PROOF OF LEMMA 3.1

*Proof.*

$$\log \mathbb{E}_\pi \left[ \sum_{t=0}^{\infty} \gamma^t r(s_t, a_t) \right] \overset{(a)}{=} \log \frac{1}{1 - \gamma} \iint p(H) p(\tau \mid H = H) r(s_H, a_H) d\tau dH$$

$$= \log \iint p(H) \frac{p(\tau \mid H = H)}{q_\theta(\tau \mid H = H)} q_\theta(\tau \mid H = H) r(s_H, a_H) d\tau dH - \log(1 - \gamma)$$

$$\overset{(b)}{\geq} \int p(H) \left( \log \int \frac{p(\tau \mid H = H)}{q_\theta(\tau \mid H = H)} q_\theta(\tau \mid H = H) r(s_H, a_H) d\tau \right) dH - \log(1 - \gamma)$$

$$\overset{(c)}{\geq} \iint p(H) q_\theta(\tau \mid H = H) \left( \log p(\tau \mid H = H) - \log q_\theta(\tau \mid H = H) + \log r(s_H, a_H) \right) d\tau dH - \log(1 - \gamma)$$

$$\overset{(d)}{=} \iint p(H) q_\theta(\tau \mid H = H) \left( \left( \sum_{t=0}^{H} \log p(s_{t+1} \mid s_t, a_t) + \underline{\log \pi_\theta(a_t \mid s_t)} - \log q_\theta(s_{t+1} \mid s_t, a_t) - \underline{\log \pi_\theta(a_t \mid s_t)} \right) + \log r(s_H, a_H) \right) d\tau dH - \log(1 - \gamma)$$

$$\overset{(e)}{=} \iint p(H) q_\theta(\tau \mid H = \infty) \left( \left( \sum_{t=0}^{H} \log p(s_{t+1} \mid s_t, a_t) - \log q_\theta(s_{t+1} \mid s_t, a_t) \right) + \log r(s_H, a_H) \right) d\tau dH - \log(1 - \gamma)$$

$$\overset{(f)}{=} \int q_\theta(\tau) \int p(H) \left( \left( \sum_{t=0}^{H} \log p(s_{t+1} \mid s_t, a_t) - \log q_\theta(s_{t+1} \mid s_t, a_t) \right) + \log r(s_H, a_H) \right) dH d\tau - \log(1 - \gamma)$$

$$\overset{(g)}{=} \int q_\theta(\tau) \mathbb{E}_{p(H)} \left[ \left( \sum_{t=0}^{H} \log p(s_{t+1} \mid s_t, a_t) - \log q_\theta(s_{t+1} \mid s_t, a_t) \right) + \log r(s_H, a_H) \right] d\tau - \log(1 - \gamma)$$

$$\overset{(h)}{=} \int q_\theta(\tau) \sum_{t=0}^{\infty} \gamma^t \left( \log p(s_{t+1} \mid s_t, a_t) - \log q_\theta(s_{t+1} \mid s_t, a_t) + (1 - \gamma) \log r(s_H, a_H) \right) d\tau - \log(1 - \gamma)$$

$$\overset{(i)}{=} \mathbb{E}_{q_\theta(\tau)} \left[ \sum_{t=0}^{\infty} \gamma^t \left( \log p(s_{t+1} \mid s_t, a_t) - \log q_\theta(s_{t+1} \mid s_t, a_t) + (1 - \gamma) \log r(s_H, a_H) - (1 - \gamma) \log(1 - \gamma) \right) \right].$$

For *(a)*, we applied Lemma A.2. For *(b)*, we applied Jensen's inequality. For *(c)*, we applied Jensen's inequality again. For *(d)*, we substituted the definitions of $p_\theta(\tau \mid H)$ and $q_\theta(\tau \mid H)$. For *(e)*, we noted that the term inside the summation only depends on the first $H$ steps of the trajectory, so collecting longer trajectories will not change the result. This allows us to rewrite the integral as an expectation using a single infinite-length trajectory. For *(f)*, we recalled the definition $q_\theta(\tau) = q_\theta(\tau = H = \infty)$ and swap the order of integration. For *(g)*, we express the inner integral over $p(H)$ as an expectation. For *(h)*, we applied the identity from Lemma A.1. For *(i)*, we moved the constant $\log(1 - \gamma)$ back inside the integral and rewrote the integral as an expectation. We have thus obtained the desired result. $\square$

### A.4 PROOF OF LEMMA 3.2

Before presenting the proof of Lemma 3.1 itself, we show how we derived the lower bound in this more general case. While this step is not required for the proof, we include it because it sheds light on how similar lower bounds might be derived for other problems. We define $\gamma_\theta(H)$ to be a learned distribution over horizons $H$. We then proceed, following many of the same steps as for the proof of

Lemma 3.1.

$$\log \mathbb{E}_\pi \left[ \sum_{t=0}^\infty \gamma^t r(s_t, a_t) \right] \overset{(a)}{=} \log \iint \frac{p(\tau, H)}{q_\theta(\tau, H)} q_\theta(\tau, H) r(s_H, a_H) d\tau dH - \log(1 - \gamma)$$

$$\overset{(b)}{\geq} \iint q_\theta(\tau, H) \left( \log p(\tau, H) - \log q_\theta(\tau, H) + \log r(s_H, a_H) d\tau \right) dH - \log(1 - \gamma) \tag{13}$$

$$\overset{(c)}{=} \int \sum_{H=0}^\infty \gamma_\theta(H) q_\theta(\tau \mid H) \left( \left( \sum_{t=0}^H \log p(s_{t+1} \mid s_t, a_t) - \log q_\theta(s_{t+1} \mid s_t, a_t) \right) + \log p(H) - \log \gamma_\theta(H) + \log r(s_H, a_H) d\tau \right) - \log(1 - \gamma)$$

$$\overset{(d)}{=} \int q_\theta(\tau \mid H = \infty) \sum_{H=0}^\infty \gamma_\theta(H) \left( \left( \sum_{t=0}^H \log p(s_{t+1} \mid s_t, a_t) - \log q_\theta(s_{t+1} \mid s_t, a_t) \right) + \log p(H) - \log \gamma_\theta(H) + \log r(s_H, a_H) d\tau \right) - \log(1 - \gamma)$$

$$\overset{(e)}{=} \int q_\theta(\tau) \sum_{H=0}^\infty \gamma_\theta(H) \left( \left( \sum_{t=0}^H \log p(s_{t+1} \mid s_t, a_t) - \log q_\theta(s_{t+1} \mid s_t, a_t) \right) + \log p(H) - \log \gamma_\theta(H) + \log r(s_H, a_H) d\tau \right) - \log(1 - \gamma)$$

$$\overset{(f)}{=} \mathbb{E}_{q_\theta(\tau)} \left[ \sum_{H=0}^\infty \gamma_\theta(H) \left( \left( \sum_{t=0}^H \log p(s_{t+1} \mid s_t, a_t) - \log q_\theta(s_{t+1} \mid s_t, a_t) \right) + \log(1 - \gamma) + H \log \gamma - \log \gamma_\theta(H) + \log r(s_H, a_H) \right) \right] - \log(1 - \gamma)$$

$$\overset{(g)}{=} \mathbb{E}_{q_\theta(\tau)} \left[ \sum_{H=0}^\infty \left( \sum_{t=H}^\infty q(t) \right) (\log p(s_{H+1} \mid s_H, a_H) - \log q_\theta(s_{H+1} \mid s_H, a_H)) + \gamma_\theta(H) (H \log \gamma - \log \gamma_\theta(H) + \log r(s_H, a_H)) \right]$$

$$\overset{(h)}{=} \mathbb{E}_{q_\theta(\tau)} \left[ \sum_{H=0}^\infty \left( 1 - \sum_{t=0}^{H-1} q(t) \right) (\log p(s_{H+1} \mid s_H, a_H) - \log q_\theta(s_{H+1} \mid s_H, a_H)) + \gamma_\theta(H) (H \log \gamma - \log \gamma_\theta(H) + \log r(s_H, a_H)) \right]$$

$$\overset{(i)}{=} \mathbb{E}_{q_\theta(\tau)} \left[ \sum_{H=0}^\infty (1 - \Gamma_\theta(H-1)) (\log p(s_{H+1} \mid s_H, a_H) - \log q_\theta(s_{H+1} \mid s_H, a_H)) + \gamma_\theta(H) (H \log \gamma - \log \gamma_\theta(H) + \log r(s_H, a_H)) \right]$$

$$\overset{(j)}{=} \mathbb{E}_{q_\theta(\tau)} \left[ \sum_{H=0}^\infty \gamma_\theta(H) \left( \frac{1 - \Gamma_\theta(H-1)}{\gamma_\theta(H)} (\log p(s_{H+1} \mid s_H, a_H) - \log q_\theta(s_{H+1} \mid s_H, a_H)) + H \log \gamma - \log \gamma_\theta(H) + \log r(s_H, a_H) \right) \right].$$

For *(a)*, we applied Lemma A.2 and multiplied the integrand by $\frac{q_\theta(\tau \mid H = H) \gamma_\theta(H)}{q_\theta(\tau \mid H = H) \gamma_\theta(H)} = 1$. For *(b)*, we applied Jensen's inequality. For *(c)*, we factored $p(\tau, H) = p(\tau, H) p(H)$ and $q_\theta(\tau, H) = q(\tau \mid H) \gamma_\theta(H)$. Note that under the joint distribution $p(\tau, H)$, the horizon $H \sim p(H) = \text{GEOM}(1 - \gamma)$ is independent of the trajectory, $\tau$. For *(d)*, we rewrote the expectation as an expectation over a single infinite-length trajectory and simplified the summand. For *(e)*, we recall the definition $q_\theta(\tau) = q_\theta(\tau = H = \infty)$. For *(f)*, we rewrote the integral as an expectation and wrote out the definition of the geometric distribution, $p(H)$. For *(g)*, we regrouped the difference of dynamics terms. For *(h)*, we noted used the fact that $\sum_{t=0}^{H-1} \gamma_\theta(t) + \gamma_{t=H}^\infty \gamma_\theta(t) = 1$. For *(i)*, we substituted the definition of the CDF function. For *(j)*, we rearranged terms so that all were multiplied by the discount $\gamma_\theta(H)$. Thus, we have obtained the desired result. We now prove Lemma 3.2, showing that Eq. 4 becomes tight at optimality.

*Proof.*

$$\mathcal{L}_\gamma(\theta) \overset{(a)}{=} \iint q_\theta(\tau, H) \left( \log p(\tau, H) - \log q_\theta(\tau, H) + \log r(s_H, a_H) d\tau \right) dH - \log(1 - \gamma)$$

$$\overset{(b)}{=} \iint q_\theta(\tau) \gamma_\theta(H \mid \tau) \left( \log p(\tau) + \log p(H) - \log q_\theta(\tau) - \log \gamma_\theta(H \mid \tau) + \log r(s_H, a_H) d\tau \right) dH - \log(1 - \gamma) \tag{14}$$

For *(a)*, we undo some of the simplifications above, going back to Eq. 13 For *(b)*, we factor $q_\theta(\tau, H) = q_\theta(\tau) \gamma_\theta(H \mid \tau)$ and $p(\tau, H) = p(\tau) p(H)$. At this point, we can solve analytically for the optimal discount distribution, $\gamma_\theta(H \mid \tau)$:

$$\gamma_\theta^*(H \mid \tau) = \frac{p(H) r(s_H, a_H)}{\sum_{H'=0}^\infty p(H') r(s_{H'}, a_{H'})} = \frac{p(H) r(s_H, a_H)}{(1 - \gamma) R(\tau)} \tag{15}$$

In the second equality, we substitute the definition of $R(\tau)$. We then substitute Eq. 15 into our expression for $\mathcal{L}_\gamma(\theta)$ and simplify the resulting expression.

$$\mathcal{L}_\gamma(\theta) = \iint q_\theta(\tau) \gamma_\theta(H \mid \tau) \left( \log p(\tau) + \log p(H) - \log q_\theta(\tau) - \log p(H) - \log r(s_H, a_H) + \log(1 - \gamma) + \log R(\tau) + \log r(s_H, a_H) d\tau \right) dH - \log(1 - \gamma)$$

$$= \iint q_\theta(\tau) \gamma_\theta(H \mid \tau) \left( \log p(\tau) - \log q_\theta(\tau) + \log R(\tau) d\tau \right) dH$$

$$= \int q_\theta(\tau) \left( \log p(\tau) - \log q_\theta(\tau) + \log R(\tau) \right) d\tau.$$

In the final line we have removed the integral over $H$ because none of the integrands depend on $H$. At this point, we can solve analytically for the optimal trajectory distribution, $q_\theta(\tau)$:

$$q^*(\tau) = \frac{p(\tau)R(\tau)}{\int p(\tau')R(\tau')d\tau'}. \tag{16}$$

We then substitute Eq. 16 into our expression for $\mathcal{L}_\gamma(\theta)$, and simplify the resulting expression:

$$\mathcal{L}_\gamma(\theta) = \int q_\theta(\tau)\left(\log p(\tau) - \log p(\tau) - \log R(\tau) + \log\int p(\tau')R(\tau')d\tau' + \log R(\tau)\right)d\tau$$

$$= \log\int p(\tau)R(\tau)d\tau = \log\mathbb{E}_\pi\left[\sum_{t=0}^{\infty}\gamma^t r(s_t, a_t)\right].$$

We have thus shown that the lower bound $\mathcal{L}_\gamma$ becomes tight when we use the optimal distribution over trajectories $q_\theta(\tau)$ and optimal learned discount $\gamma_\theta(H \mid \tau)$. $\qquad\square$

### A.5 A LOWER BOUND FOR GOAL-REACHING TASKS.

Many RL problems can be better formulated as goal-reaching problems, a formulation that does not require defining a reward function. We now introduce a variant of our method for goal-reaching tasks. Using $\rho^\pi(s_{t+})$ to denote the discounted state occupancy measure of policy $\pi$, we define the goal-reaching objective as maximizing the probability density of reaching a desired goal $s_g$:

$$\max_\theta \log\rho^{\pi_\theta}(s_{t+} = s_g). \tag{17}$$

We refer the reader to Eysenbach et al. (2020b) for a more detailed discussion of this objective. For simplicity, we assume that the goal is fixed, noting that the multi-task setting can be handled by conditioning the policy on the commanded goal. Similar to Lemma 3.1, we can construct a lower bound on the goal-conditioned RL problem:

**Lemma A.3.** *Let initial state distribution $p_1(s_1)$, real dynamics $p(s_{t+1} \mid s_t, a_t)$, reward function $r(s_t, a_t) > 0$, discount factor $\gamma \in (0, 1)$, and goal $g$ be given. Then the following bound holds for any dynamics $q(s_{t+1} \mid s_t, a_t)$ and policy $\pi(a_t \mid s_t)$:*

$$\log p^{\pi_\theta}(s_{t+} = s_g) \geq \mathbb{E}_{q^{\pi_\theta}(\tau)}\left[\sum_{t=0}^{\infty}\gamma^t r(s_t, a_t)\right], \tag{18}$$

*where $\tilde{r}_g(s_t, a_t, s_{t+1}) \triangleq (1-\gamma)\log p(s_{t+1} = s_g \mid s_t, a_t) + \log p(s_{t+1} \mid s_t, a_t) - \log q(s_{t+1} \mid s_t, a_t)$.*

The proof, presented below, is similar to the proof of Lemma 3.1. The first term in the reward function, the log probability of reaching the commanded goal one time step in the future, is similar to prior work (Rudner et al., 2021). The correction term $\log p - \log q$ incentivizes the policy to avoid transitions where the model is inaccurate, and can be estimated using a separate classifier. One important aspect of this goal-reaching problem is that it is entirely data-driven, avoiding the need for any manually-designed reward functions.

*Proof.*

$$\log\rho^{\pi_\theta}(s_{t+} = s_g) \overset{(a)}{=} \log\iint p(H)p(\tau \mid H = H)p(s_g \mid s_H, a_H)d\tau dH$$

$$= \log\iint p(H)\frac{p(\tau \mid H = H)}{q_\theta(\tau \mid H = H)}q_\theta(\tau \mid H = H)\frac{p(s_g \mid s_H, a_H)}{q_\theta(s_g \mid s_H, a_H)}q_\theta(s_g \mid s_H, a_H)d\tau dH - \log(1-\gamma)$$

$$\overset{(c)}{\geq} \iint p(H)q_\theta(\tau \mid H = H)\left(\log p(\tau \mid H = H) - \log q_\theta(\tau \mid H = H) + \log p(s_g \mid s_H, a_H) - \log q_\theta(s_g \mid s_H, a_H)\right)d\tau dH - \log(1-\gamma)$$

$$\overset{(d)}{=} \iint p(H)q_\theta(\tau \mid H = \infty)\left(\sum_{t=0}^{H}\log p(s_{t+1} \mid s_t, a_t) - \log q_\theta(s_{t+1} \mid s_t, a_t)\right) + \log p(s_g \mid s_H, a_H) - \log q_\theta(s_g \mid s_H, a_H)d\tau dH - \log(1-\gamma)$$

$$\overset{(d)}{=} \int q_\theta(\tau)\int p(H)\left(\sum_{t=0}^{H}\log p(s_{t+1} \mid s_t, a_t) - \log q_\theta(s_{t+1} \mid s_t, a_t)\right) + \log p(s_g \mid s_H, a_H) - \log q_\theta(s_g \mid s_H, a_H)dH d\tau - \log(1-\gamma)$$

$$\overset{(d)}{=} \int q_\theta(\tau)\sum_{t=0}^{\infty}\gamma^t\left(\log p(s_{t+1} \mid s_t, a_t) - \log q_\theta(s_{t+1} \mid s_t, a_t) + (1-\gamma)(\log p(s_g \mid s_t, a_t) - \log q_\theta(s_g \mid s_t, a_t))\right)d\tau - \log(1-\gamma)$$

$$\overset{(d)}{=} \mathbb{E}_{q_\theta(\tau)}\left[\sum_{t=0}^{\infty}\gamma^t\left(\log p(s_{t+1} \mid s_t, a_t) - \log q_\theta(s_{t+1} \mid s_t, a_t) + (1-\gamma)(\log p(s_g \mid s_t, a_t) - \log q_\theta(s_g \mid s_t, a_t) + (1-\gamma)\log(1-\gamma))\right)\right].$$

□

Similar to the more complex lower bound presented in Eq. 4, this lower bound on goal-reaching can be modified (by learning a discount factor) to become a tight lower bound. The resulting objective would resemble a model-based version of the algorithm from Rudner et al. (2021).

### A.6 DERIVATION OF MODEL OBJECTIVE (EQ. 10)

Our lower bound depends on entirely trajectories sampled from the learned dynamics. In this section, we show how the same objective can be expressed as an expectation of transitions. This expression is easier to optimize, as it does not require backpropagating gradients through time. We start by writing our lower bound, conditioned on a current state $s_t$.

$$\mathbb{E}_{\substack{\pi(a_t|s_t),\\ q_\theta(s_{t+1}|s_t,a_t)}} \left[ \sum_{t'=t}^{\infty} \gamma^{t'-t} \widetilde{r}(s_{t'}, a_{t'}) \mid s_t \right]$$

$$= \mathbb{E}_{\substack{\pi(a_t|s_t),\\ q_\theta(s_{t+1}|s_t,a_t)}} \left[ \widetilde{r}(s_t, a_t, s_{t+1}) + \gamma V(s_{t+1}) \mid s_t \right]$$

$$\overset{(a)}{=} \mathbb{E}_{\substack{\pi(a_t|s_t),\\ q_\theta(s_{t+1}|s_t,a_t)}} \left[ (1-\gamma)\log r(s_t, a_t) + \log \frac{C_\phi(s_t, a_t, s_{t+1})}{1 - C_\phi(s_t, a_t, s_{t+1})} - (1-\gamma)\log(1-\gamma) + \gamma V(s_{t+1}) \mid s_t \right]$$

In *(a)*, we substituted the definition of the augmented return. For the purpose of optimizing the dynamics model, we can ignore all terms that do not depend on $s_{t+1}$. Removing these terms, we arrive at our model training objective (Eq. 10)

## B ADDITIONAL EXPERIMENTS

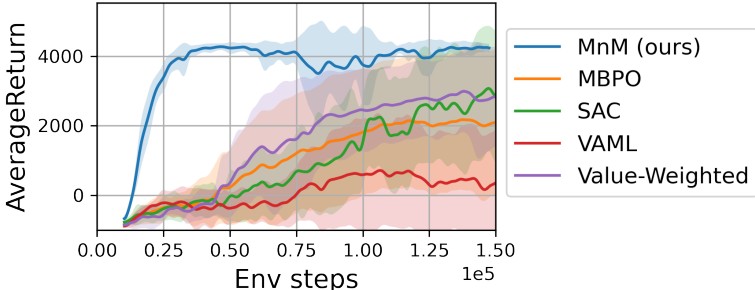

Figure 8: **Alternative Model Learning Objectives**: Using the `DClawScrewFixed-v0` task, we compare MnM and MBPO (Janner et al., 2019) to two additional model learning objectives suggested in the literature, VAML (Farahmand et al., 2017) and value-weighted maximum likelihood (Lambert et al., 2020). MnM (our method) outperforms these alternative approaches.

We compare MnM to a number of alternative model learning methods. MBPO (Janner et al., 2019) uses a standard maximum likelihood model. We implement a version of VAML (Farahmand et al., 2017), which augments the maximum likelihood loss with an additional temporal difference loss; the model should predict next states that have low Bellman error. Finally, we compare to a variant of the MBPO maximum likelihood model that weights transitions based on the Q values, an idea discussed (but not actually implemented) in Lambert et al. (2020). We implement this value weighting method by computing the Q values for the current states and computing a softmax over the batch dimension to obtain per-example weights.

We use the `DClawScrewFixed-v0` task for this experiment. The results, shown in Fig. 8, show that MnM outperforms these alternative approaches. We observe that the value-weighting performs

slightly better than the standard maximum likelihood model, while the VAML method performs noticeable worse.

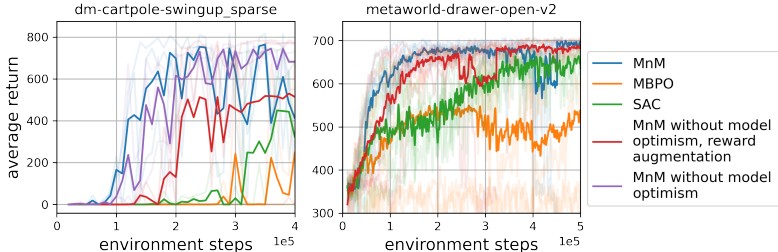

Figure 9: **Ablations Experiments**: Compared with MBPO (orange line), MnM uses a GAN-like model (red line) with a model optimism term and modifies the reward function.

We next run an ablation experiment to study the importance of a few key design decisions. We compare MnM with ablations that omit the reward augmentation and the model optimism term. The results shown in Fig. 9 indicate that most of the benefit of MnM comes from using a GAN-like model. Because the dynamics of these tasks are nearly deterministic, it is not surprising that the optimistic dynamics and the reward augmentation have only a small effect.

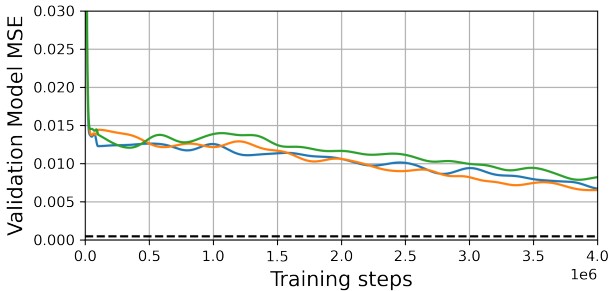

Figure 10: **MnM trains stably.** Despite resembling a GAN, the MnM dynamics model trains stably, with the validation MSE decreasing steadily throughout training. Different colors correspond to different random seeds of MnM. The dashed line corresponds to the minimum validation MSE of a maximum likelihood dynamics model.

With the implementation details described in Appendix C, we found that the MnM dynamics model trained stably, despite resembling a GAN. In Fig. 10, we plot the validation MSE of the MnM model throughout training, observing that it decreases monotonically. Different lines correspond to different random seeds, and the dashed line corresponds to the minimum MSE of a maximum likelihood model (a MBPO model). Note that the MnM model is not trained with this MSE objective, but with the GAN-like objective in Eq. 10. It is therefore not surprising that MnM does not perform as well on this objective as the maximum likelihood model.

## C  IMPLEMENTATION DETAILS

All experiments were run on at least three random seeds.

### C.1  ALGORITHMS

**Value Iteration (Fig. 2a *(right)*, Fig. 2b, Fig. 3)**    For the tabular experiments that perform value iteration, we perform Polyak averaging of the policy and learned dynamics model with parameter $\tau = 0.5$. We found that the value iteration version of MnM diverged without this Polyak averaging step. Experiments were stopped when the MnM dynamics model (which depends on the value function) changed by less than 1e-6 across iterations, as measured using an $L_0$ norm. For VMBPO we used $\eta = 1$.

|  | SAC | MBPO | MnM |
|---|---|---|---|
| HalfCheetah-v2 | - | 40 | 20 |
| Hopper-v2 | - | 40 | 20 |
| Walker2d-v2 | - | 40 | 20 |
| dm-cartpole-swingup_sparse | 1 | 1 | 1 |
| metaworld-window-open-v2 | 20 | 20 | 20 |
| metaworld-door-open-v2 | 40 | 40 | 40 |
| metaworld-reach-v2 | 20 | 20 | 20 |
| metaworld-drawer-open-v2 | 40 | 40 | 40 |
| DClawPoseRandom-v0 | 20 | 20 | 20 |
| DClawTurnRandom-v0 | 40 | 40 | 40 |
| DClawScrewFixed-v0 | 40 | 40 | 40 |
| DClawScrewRandom-v0 | 40 | 40 | 40 |

Table 1: **Gradient updates per real environment step**: This parameter was separately tuned for each method and each environment.

**Q-learning (Fig. 2)**   For the experiments with Q-learning (both with and without the MnM components), we performed $\epsilon$-greedy exploration with $\epsilon = 0.5$. We used a learning rate of $1e-2$. For this task alone, we compute the MnM dynamics analytically by combining the true environment dynamics with the learned value function, allowing for clearer theoretical analysis. For fair comparison, all methods receive the same amount of data, perform the same number of updates, and are evaluated using the real environment dynamics. For VMBPO we used $\eta = 1$.

**SAC for continuous control tasks.**   We used the SAC implementation from TF-Agents (Guadarrama et al., 2018) with the default hyperparameters.

**MBPO for continuous control tasks.**   We implement MBPO on top of the SAC implementation from TF-Agents (Guadarrama et al., 2018). Unless otherwise mentioned, we take the default parameters from this implementation. We use an ensemble of 5 dynamics models, each with 4 hidden layers of size 256. The dynamics model predicts the whitened difference between the next state and the current state. That is, to obtain the prediction for the next state, the predictions are scaled by a per-coordinate variance, shifted by a per-coordinate mean, and then added to the current state. These whitened predictions are clipped to have a minimum standard deviation of 1e-5; without this, we found that the MBPO model resulted in numerical instability. The model is trained using the standard maximum likelihood objective, with all members of the ensemble being trained on the same data. To sample data from this model we perform 1-step rollouts, starting at states visited in the true dynamics. We perform one batch of rollouts in parallel using a batch size of 256. To sample the corresponding action, with probability 50% we take the action that was executed in the true dynamics; with probability 50% we sample an action from the current policy. We found that this modification slightly improves the results of MBPO. We use a batch size of 256. We have two replay buffers: the model replay buffer has size 256e3 and the replay buffer of real experience has size 1e6. At the start of training, we collect 1e4 transitions from the real environment, train the dynamics model on this experience for 1e5 batches, and only then start training the policy. We use a learning rate of 3e-4 for all components. To stabilize learning, we maintain a target dynamics model using an exponential moving average ($\tau = 0.001$), and use this target dynamics model to sample transitions for training. We update the model, policy, and value functions at the same rate we sample experience from the learned model, which is more frequently than we collect experience from the real environment (see Table 1).

**MnM for Continuous Control Tasks**   We implement MnM on top of the SAC implementation from TF-Agents (Guadarrama et al., 2018). Unless otherwise mentioned, we take the default parameters from this implementation. Our model architecture is exactly the same as our MBPO implementation, and we follow the same training protocol.

Unlike MBPO, MnM also learns a classifier for distinguishing real versus model transitions. The classifier architecture is a 2 layers neural network with 1024 hidden units in each layer. We found that this large capacity was important for stable learning. We add input noise with $\sigma = 0.1$ while

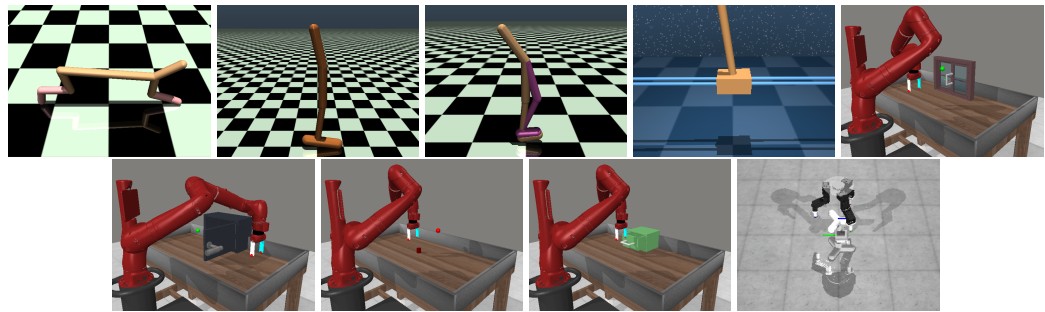

Figure 11: **Environments**: Our experiments look at three locomotion tasks from OpenAI Gym (Brockman et al., 2016), the inverted pendulum task from DM Control (Tassa et al., 2018), four manipulation tasks from Metaworld (Yu et al., 2020a), and four dextrous manipulation tasks from Robel (Ahn et al., 2020). The Robel tasks use the same dynamics but different reward functions, so we only include an image of one task.

training the classifier. We whiten the inputs to the classifier by subtracting a coordinate-wise mean and dividing by a coordinate-wise standard deviation. When training the classifier, we take samples from both the dynamics model and the target dynamics model as negative examples, finding that this stabilizes learning somewhat. Following the suggestion of prior work (Salimans et al., 2016), we use one-sided label smoothing with value 0.1, only smoothing the negative predictions and not the positive predictions.

We found that gradient penalties and spectral normalization decreased performance. We found that automatically tuning the classifier input noise also decreased performance. We found that mixup had little effect. We found that the loss would often plateau around 1e4 batches, but would eventually start decreasing again after 2e4 - 2e5 batches.

Like the MBPO model, we first collect 1e4 transitions of experience from the real environment using a random policy, then train the dynamics model and classifier for 1e5 batches, and only then start updating the policy. Because the Q values are poor at the start of training, we only add the value term to the model loss (resulting in the optimistic dynamics model) after 2e5 batches (1e5 batches of model training, then 1e5 batches of model+policy training). To further improve stability, we compute the value term in the model loss by taking the minimum over two target value functions (like TD3 (Fujimoto et al., 2018)). We update the model, classifier policy, and value functions at the same rate we sample experience from the learned model, which is more frequently than we collect experience from the real environment (see Table 1).

## C.2 Environments

**Gridworld for Fig. 2a.** This task is a $10 \times 10$ gridworld with obstacles shown in Fig. 2a. There are four discrete actions, corresponding to moving to the four adjacent cells. With probability 50%, the agent's action is ignored and a random action is taken instead. The agent starts in the top-left cell. The agent receives a reward of +0.001 at each time step and a reward of +10 when at the goal state. Episodes have 200 steps and we use a discount $\gamma = 0.9$.

**Gridworld for Fig. 2b.** This task is a $15 \times 15$ gridworld with obstacles shown in Fig. 2b. There are four discrete actions, corresponding to moving to the four adjacent cells. The dynamics are deterministic. The agent starts in the top-left cell. The reward is +1.0 at every state except the goal state, where the agent receives a reward of +100.0. We use $\gamma = 0.9$ and compute optimal policies analytically using value iteration. We implementing the aliasing by averaging together the dynamics for each block of $3 \times 3$ states. Importantly, the averaging was done to the *relative* dynamics (e.g., action 1 corresponds to move right) not the *absolute* dynamics (e.g., action 1 corresponds to moving to state (3, 4)). To handle edge effects, we modified the averaged dynamics so that the agent could not exit the gridworld. We computed the classifier analytically using the true and learned dynamics models. However, the augmented rewards become infinite because the learned model assigns non-zero probability to transitions that cannot occur under the true dynamics. This is not a failure of our theory, as the optimal policy would choose to never visit these states, but it presents a challenge for optimization. We therefore added label smoothing with parameter 0.7 to the classifier.

**Gridworld for Fig. 3a.** This task is a $10 \times 10$ gridworld with obstacles shown in Fig. 2. There are four discrete actions, corresponding to moving to the four adjacent cells. With probability 90%, the agent's action is ignored and a random action is taken instead. The agent starts in the top-left cell. The rewards depend on the Manhattan distance to the goal: transitions that lead away from the goal have a reward of +0.001, transitions that do not change the distance to the goal have a reward of +1.001, and transitions that decrease the distance to the goal have a reward of +2.001. We use $\gamma = 0.5$ and compute values and returns analytically using value iteration.

**Gridworld for Fig. 3b.** This task used the same dynamics as Fig. 2a. The one change is that the agent receives a reward of +1.0 at each time step and a reward of +10.0 at the goal state. We estimate the more complex lower bound (Eq. 4) by also learning the discount factor. However, since we currently do not have a method for learning non-Markovian dynamics to fully optimize this lower bound, we do not expect the lower bound to become tight.

**`HalfCheetah-v2, Hopper-v2, Walker2d-v2.`** These tasks are taken directly from the OpenAI benchmark (Brockman et al., 2016) without modification.

**`dm-cartpole-swingup_sparse.`** This task is taken directly from the DM Control benchmark (Tassa et al., 2018) without modification.

**`metaworld-window-open-v2.`** This task is based on the `window-open-v2` task from the Metaworld benchmark (Yu et al., 2020a). To increase the difficulty of this task, we set `random_init=True` and use a sparse reward function. The sparse reward function is +0 when the window is within 3 units of the (default) open position and -1 otherwise.

**`metaworld-door-open-v2.`** This task is based on the `door-open-v2` task from the Metaworld benchmark (Yu et al., 2020a). To increase the difficulty of this task, we set `random_init=True` and use a sparse reward function. The sparse reward function is +1 when the window is within 3 units of the (default) open position and +0 otherwise.

**`metaworld-reach-v2.`** This task is based on the `reach-v2` task from the Metaworld benchmark (Yu et al., 2020a). To increase the difficulty of this task, we set `random_init=True` and use a sparse reward function. The sparse reward function is +0 when the window is within 1 units of the goal and -1 otherwise.

**`metaworld-drawer-open-v2.`** This task is based on the `drawer-open-v2` task from the Metaworld benchmark (Yu et al., 2020a). To increase the difficulty of this task, we remove the reward shaping term (`reward_for_caging`) and just optimize the reward for opening the drawer (`reward_for_opening`).

**`DClawPoseRandom-v0, DClawTurnRandom-v0, DClawScrewFixed-v0, DClawScrewRandom-v0.`** These tasks are taken directly from the ROBEL benchmark (Ahn et al., 2020) without modification.

