# OpenReview forum: "Mismatched No More: Joint Model-Policy Optimization for Model-Based RL"
_ICLR.cc/2022/Conference — ICLR 2022 Submitted_

### Official Review · Reviewer_yCzC · 2021-11-02

**Correctness:** 4
**Technical Novelty And Significance:** 3
**Empirical Novelty And Significance:** 3
**Recommendation:** 6
**Confidence:** 4

**Main Review:**

The method is well motivated, and the proposed augmented reward function is novel. The experiments are also well conducted, and the visualization of the learned dynamics provides intuitive illustrations that it tends to help the policy to succeed. The overall performance of MnM is also demonstrated to perform at least as well compared with SAC and MBPO.

Despite these advantages mentioned above, I have some comments below that I’d like to hear from the authors in the responses.

1. The augmented reward in Eq. (3) is informative, while in Theorem 3.1, the equality of the lower bound is hard to hold. In the proof of Theorem 3.1, Jensen’s inequality is applied, twice, to move the logarithm inside the expectation. For Jensen’s inequality, the equality holds when the function (logarithm here) is affine or the variable inside the function (the reward r here) is constant. Obviously, neither is the case here. Therefore, at least for the reward function in Eq. (3), maximizing over its cumulative expectation does not guarantee the optimality in the original expected return.

2. Although, immediately, the following contexts about “Tightening the lower bound” gives a new reward function by employing a learnable $\gamma_{\theta}$, I cannot agree that this is really significant. From the proof of Lemma 3.2, the equality holds when the two analytical solutions are satisfied for $\gamma_{\theta}$ and the dynamics model, and the two analytical solutions are independent with each other. Then, for the learnable $\gamma_{\theta}$, it indeed plays a role of reward shaping that shapes the reward in Eq. (3) probably nonlinearly to reach the logarithm of the original expected return (probably a way to approximate the affine function to remove the Jensen’s gap?). Also, in the experiments, this new lower bound and reward function are not applied. So, I agree Lemma 3.2 demonstrates that there exists some way to modify the reward in Eq. (3) to have a tighter lower bound, while I think this is a benefit from reward shaping, instead of a thoroughly informative new reward function.

Minor:
In the proof of Lemma 3.1, line (d) in the equation, the reduced $\pi_{\theta}(a_t|s_t)$ missed the logarithm.

Overall, I think the proposed method is novel and the experiments give nice explanations for the learned dynamics in tending to assist the policy learning, and MnM indeed has its superiority compared to prior model-based RL methods.


**Summary Of The Paper:**

This paper proposes a new model-based RL method by devising a new reward function, which incorporates a term measuring the difference between the learned dynamics model and the true dynamics probability, in addition to the logarithm of the original environment reward. Maximizing this augmented reward function is proved to maximizing a lower bound of the logarithm of the original expected return. The reward function is not practically usable since the true dynamics probability is not known. To practically apply the proposed reward function, a GAN-like classifier is introduced to differentiate the transition generated from the dynamics model and the true dynamics. The experiments show that under an unified objective, the learned dynamics tends to assign higher transition probability for some (s,a,s’) if this transition can assist the policy to achieve a higher return in the true environments.

**Summary Of The Review:**

A good paper that introduces a novel model-based RL method, where the policy learning and dynamics modeling share a unified objective.

---

> ### Author Response · Authors · 2021-11-18
> **Response to yCzC**
>
> We thank the reviewer for their detailed feedback! The reviewer's main concern seems to be about clarifying the explanation of the lower bound (e.g., emphasizing that the simple lower bound does not always hold with equality). We have revised the paper to address this issue. **Are there additional revisions that the reviewer would like to see?** If so, we would be happy to update the paper.
>
> > In Theorem 3.1, the equality of the lower bound is hard to hold.
>
> We believe that the statement of Theorem 3.1 is correct: the proposed objective is either less than or equal to the expected return objective. We agree with the reviewer that this lower bound will not, in general, hold with equality. In cases where the bound is loose, we are guaranteed that the policy that optimizes the lower bound will also get high reward on the true environment, but it remains an open question whether it will get the highest reward on the true environment. Nonetheless, we find empirically that maximizing this simple lower bound does result in policies that get high return on a wide range of tasks. To clarify that we are optimizing a potentially-loose lower bound, we have revised the final paragraph in "An objective for model-based RL."
>
> > “Tightening the lower bound” gives a new reward function..., I cannot agree that this is really significant.
>
> We believe that the objective in Eq. 4 may be of *theoretical* interest because it combines properties that current model-based RL objectives lack:
> * At optimality, the objective is equal to the expected return objective.
> * The model and policy are optimized using the same objective. This makes it easy to determine which models are better for the purpose of policy optimization (similar to [1, 2, 3]).
> * The objective is a global lower bound on expected return. While prior work [4] has proposed a *local* lower bound, a *global* lower bound is arguably much more useful, as it holds for any choice of model and policy.
> We have added a paragraph to Section 2 to explain these comparisons with prior work.
>
> We agree that the *practical* significance of this theoretical result is currently limited, as we have not proposed a method for optimizing this objective (last paragraph of "Tightening the lower bound.").
>
> > missing "log" in line (d) of proof of Lemma 3.1
>
> We have fixed this typo.
>
> [1] Grimm, Christopher, André Barreto, Satinder Singh, and David Silver. "The value equivalence principle for model-based reinforcement learning." arXiv preprint arXiv:2011.03506 (2020).
>
> [2] Nikishin, Evgenii, Romina Abachi, Rishabh Agarwal, and Pierre-Luc Bacon. "Control-Oriented Model-Based Reinforcement Learning with Implicit Differentiation." arXiv preprint arXiv:2106.03273 (2021).
>
> [3] D'Oro, Pierluca, Alberto Maria Metelli, Andrea Tirinzoni, Matteo Papini, and Marcello Restelli. "Gradient-aware model-based policy search." In Proceedings of the AAAI Conference on Artificial Intelligence, vol. 34, no. 04, pp. 3801-3808. 2020.
>
> [4] Luo, Yuping, et al. "Algorithmic framework for model-based deep reinforcement learning with theoretical guarantees." arXiv preprint arXiv:1807.03858 (2018).

---

> > ### Author Response · Authors · 2021-11-22
> > **Response to yCzC**
> >
> > Dear Reviewer,
> >
> > We hope that you've had a chance to read our response. We would really appreciate a reply as to whether our response and clarifications have addressed the issues raised in your review, or whether there is anything else we can address.

---

> > > ### Comment · Reviewer_yCzC · 2021-11-24
> > > **Response to authors**
> > >
> > > Dear authors,
> > >
> > > Thank you for your reply. I think the changes in the revision are more accurate to position the theoretical results. I do not have further questions.

---

### Official Review · Reviewer_uKVv · 2021-11-02

**Correctness:** 4
**Technical Novelty And Significance:** 2
**Empirical Novelty And Significance:** 2
**Recommendation:** 6
**Confidence:** 3

**Main Review:**

Objective mismatch problem is important in MBRL. And the proposed MnM is motivated by the idea of GAN to solve this problem. The whole paper is easy to follow and I also appreciate the theorems in the paper. However, I would like to see more comparisons, both theoretical and experimental, that if MnM performs better than the VAML framework [1,2] and the simple weighting strategy in [3]. Both these works are addressing the same problem, so it would be important to directly compare them, especially considering that MnM is a more complicated algorithm.

[1] Value-Aware Loss Function for Model-based Reinforcement Learning. Amir-massoud Farahmand et al. AISTATS 2017\
[2] Iterative Value-Aware Model Learning. Amir-massoud Farahmand et al. NeurIPS 2018 \
[3] Objective Mismatch in Model-based Reinforcement Learning. Nathan Lambert et al. L4DC 2020

**Summary Of The Paper:**

This paper concerns the objective mismatch problem in model-based RL that the model is optimized for prediction accuracy, not a good performance. The authors propose Mismatched no More (MnM) that maximizes a lower bound on expected reward in an adversarial training manner. They also give a global lower bound that holds for any dynamics. Experiments also validate the effectiveness of MnM.

**Summary Of The Review:**

The overall paper is motivated and clear, The concern I have is the advantage over previous works that address exactly the same problem with relatively simple mechanisms.

---

> ### Author Response · Authors · 2021-11-18
> **Response to uKVv**
>
> We thank the reviewer for their feedback. The reviewer's sole concern seemed to be the comparison with VAML and the simple weighting strategy proposed in (Lambert 2020). We have run this additional experiment and included results in Figure 8.  We use the DClawScrewFixed-v0 task for this comparison because its high dimensionality (36-dimensional observation) and complex contact dynamics are likely to cause model exploitation. MnM outperforms these alternative model learning approaches. **Does this new experiment address the reviewer's concerns with the paper?**

---

> > ### Author Response · Authors · 2021-11-22
> > **Response to uKVv**
> >
> > Dear Reviewer,
> >
> > We have run the experiments suggested by the reviewer. We would really appreciate a reply as to whether our response has addressed the issues raised in your review, or whether there is anything else we can address.

---

> > > ### Comment · Reviewer_uKVv · 2021-11-23
> > > **Response to authors**
> > >
> > > Thank the authors for the additional experiments. It seems that in high-dimensional systems, the proposed algorithm experimentally outperforms previous straightforward ones. Could you give a more formal comparison between MnM and VAML/value-weighted maximum likelihood? For example why MnM is theoretically better and what properties MnM has are not held by VAML/value-weighted methods. I would like to increase my score if it's convincing. For the current version, I still can't see the obvious reasons that people will choose the complicated MnM as a tool to address objective mismatch problems instead of choosing the straightforward ones.

---

> > > > ### Author Response · Authors · 2021-11-23
> > > > **Why users might prefer MnM**
> > > >
> > > > There are two reasons why users might prefer to use MnM, compared with VAML (Farahmand 2017) and the value-weighted maximum likelihood (Lambert 2020):
> > > >
> > > > * **MnM avoids objective mismatch**: While VAML and value-weighted maximum likelihood take steps to address the objective mismatch problem, they do not avoid it: the models and policies trained by these methods are optimized using different objectives. In contrast, our method does avoid the objective mismatch problem: the model and policy are optimized using the same objective. Avoiding model mismatch is practically useful because it tells users that taking gradient steps to update the model will also improve the policy. Otherwise, updates to the model can make the policy worse; Fig. 4 from Lambert 2020 does a great job illustrating this point.
> > > > * **MnM optimizes a lower bound**: Even in the presence of model function approximation, MnM optimizes a lower bound on expected return. VAML and value-weighted maximum likelihood do not maximize a lower bound on expected return, and are liable to overestimate the policy's expected return in the real environment. Optimizing a lower bound, as done by MnM, is practically useful because it gives users confidence that an agent's performance in the real world will be at least as good as the performance under the learned model.

---

### Official Review · Reviewer_D2vZ · 2021-11-02

**Correctness:** 2
**Technical Novelty And Significance:** 2
**Empirical Novelty And Significance:** 2
**Recommendation:** 3
**Confidence:** 4

**Main Review:**

Novelty

To the best of the reviewer's knowledge, the idea is novel. The authors propose a lower bound on the logarithm of expected returns that the policy and the model can jointly optimize against an adversary that tries to distinguish between real and model rollouts. As a result, the MnM model mitigates the objective mismatch yet produces realistic samples. This contrasts the approach with other non-MLE methods (e.g. see [1]) that address the objective mismatch but produce unrealistic rollouts. The authors might want to emphasize the property more in the paper and study the effect in more detail.

Significance

The significance of the paper is limited. The main concern of the reviewer is about the empirical results:
- First, most experiments appear to have a very limited number of runs. The reviewer did not find the details in the paper explicitly articulating the number of random seeds. The reviewer might infer from the plots that MnM and the baselines were run 2-3 times (e.g. Figure 5, `meta-world-door-open-v2`). Given the high variance of the results, it is hard to make conclusions about the comparison of MnM to the baselines. Reporting the results using 10 random seeds might improve the credibility of the results. See [2] for a discussion about the significance of the experimentation in RL.
- Second, a few important baselines are missing. In tabular experiments, there is no comparison to an MLE MBRL agent. In continuous control, there is no comparison to any MBRL algorithm that addresses the objective mismatch. Why not use VMBPO in these experiments if you already used it in tabular experiments? Furthermore, Appendix A.4 suggests that the optimal dynamics for the lower bound are achieved by weighting the true dynamics with returns. Why not try return-weighted MLE as a baseline? Including other baselines that address the objective mismatch would further increase the significance of the results (see notes about related work).
- Lastly, while MnM mitigates the objective mismatch, it is unclear from the results what are the benefits of this. Figure 5 suggests that out of 12 continuous control environments, only on `DClawScrewRandom-v0` we can make a conclusion that MnM significantly outperforms the presented baselines. In 3 MuJoCo environments, the performance of MnM and MBPO does not differ significantly. In the other 8 environments, it is hard to make definite conclusions due to the variance of returns.

The authors provide theoretical analyses associated with MnM. There are several concerns about this part of the paper as well:
- The proposed method optimizes a bound $L(\theta)$ on the log expected returns. The authors claim that “this bound becomes tight under certain assumptions”. To support the claim, the paper introduces another bound $L_\gamma(\theta)$ and proves that the supremum of $L_\gamma(\theta)$ coincides with the log expected returns. While the reviewer appreciates the similarity of the $L_\gamma(\theta)$ with $L(\theta)$, the new bound relies on (a) non-markovian dynamics (b) **time-varying** and **parameterized** discount factor, limiting the justification of the method by this theoretical result.
- With the above in mind, it is possible to argue that even a standard MLE-based MBRL algorithm optimizes a lower bound on the expected returns (yet this bound will not be tight) using the Simulation Lemma [3]. See section 2 in [4] for an explanation. Since the bound that the practical algorithm optimizes is not tight, it is unclear to which extent the proposed method mitigates the objective mismatch.
- Lastly, the analyses make several assumptions that might not be entirely justified. The most noticeable one is the assumption about the positive reward. While it is common to assume that the rewards are *non-negative*, the assumption about positive rewards might be limiting. Moreover, the rewards in certain environments (e.g. `HalfCheetah-v2`) where MnM is tested take negative values.

Detailed notes and questions

There are several smaller concerns and questions that the authors might address:
1. The literature review could be improved. In particular, relevant references include [1] that characterize the set of models that are optimal for planning; [5] that optimize the policy and the model using the same objective; [6] that learn the model to directly optimize agent’s performance, not a lower bound on it; [7] that learn the model that is helpful for policy improvement using a weighting scheme.
2. Since $\Gamma_\theta(t)$ is a CDF, does it imply that $\gamma(t)$ is a distribution?
3. “One detail of note is that we omit the reward augmentation for MnM during these experiments, as it hinders exploration leading to lower returns.” Is it the entropy augmentation of SAC or the proposed augmentation with the log density ratio?
4. Is the reward function learned as well?

References

[1] Grimm, Christopher, André Barreto, Satinder Singh, and David Silver. "The value equivalence principle for model-based reinforcement learning." arXiv preprint arXiv:2011.03506 (2020).

[2] Henderson, Peter, Riashat Islam, Philip Bachman, Joelle Pineau, Doina Precup, and David Meger. "Deep reinforcement learning that matters." In Proceedings of the AAAI conference on artificial intelligence, vol. 32, no. 1. 2018.

[3] Kearns, Michael, and Satinder Singh. "Near-optimal reinforcement learning in polynomial time." Machine learning 49, no. 2 (2002): 209-232.

[4] Farahmand, Amir-massoud, Andre Barreto, and Daniel Nikovski. "Value-aware loss function for model-based reinforcement learning." In Artificial Intelligence and Statistics, pp. 1486-1494. PMLR, 2017.

[5] Schrittwieser, Julian, Ioannis Antonoglou, Thomas Hubert, Karen Simonyan, Laurent Sifre, Simon Schmitt, Arthur Guez et al. "Mastering atari, go, chess and shogi by planning with a learned model." Nature 588, no. 7839 (2020): 604-609.

[6] Nikishin, Evgenii, Romina Abachi, Rishabh Agarwal, and Pierre-Luc Bacon. "Control-Oriented Model-Based Reinforcement Learning with Implicit Differentiation." arXiv preprint arXiv:2106.03273 (2021).

[7] D'Oro, Pierluca, Alberto Maria Metelli, Andrea Tirinzoni, Matteo Papini, and Marcello Restelli. "Gradient-aware model-based policy search." In Proceedings of the AAAI Conference on Artificial Intelligence, vol. 34, no. 04, pp. 3801-3808. 2020.



**Summary Of The Paper:**

The paper studies the model learning aspect in model-based reinforcement learning (MBRL). Standard Dyna-like MBRL approaches that train the model using the maximum likelihood estimation (MLE) or its variations. In contrast, the authors propose an algorithm called Mismatched no More (MnM) that optimizes model parameters using a lower bound on the agent’s performance. By doing so, the paper mitigates the known objective mismatch of standard MBRL: MLE-optimized model might not necessarily be useful for optimizing agent’s returns, the overall objective of an RL system. Experimentally, the paper presents two studies. First, in 2 tabular environments, the authors compare MnM to Q-learning and VMBPO, an alternative MBRL algorithm that addresses the objective mismatch. Second, the authors compare MnM to an MLE-like MBRL baseline and a model-free algorithm on 12 continuous control tasks.


**Summary Of The Review:**

The reviewer recommends rejecting the paper. While the idea is promising, the submission in its current state needs substantial improvements. Addressing the outlined concerns might increase the overall score.

---

> ### Author Response · Authors · 2021-11-18
> **Response to D2vZ**
>
> **[response 1 of 3]**
>
> We thank the reviewer for their detailed review. The reviewer raised questions about both the empirical and theoretical aspects of the paper. We have added three additional baselines to our experiments and run an additional experiment increasing random seeds. We have also revised the paper to clarify our theoretical results, clarifications that we explain below. **Have the revisions to the paper addressed all the reviewer's concerns?**
>
> > This contrasts the approach with other non-MLE methods that address the objective mismatch but produce unrealistic rollouts. The authors might want to emphasize the property more in the paper and study the effect in more detail.
>
> Thank you for the suggestion. To emphasize these properties, we have added a sentence to the last paragraph in the introduction. We have also run a number of additional experiments to study the effect in more detail:
> * Figure 1: We have added VMBPO as a baseline.
> * Figure 8: We have added two new baselines: value-weighted maximum likelihood [2] and VAML [3]. These comparisons allow us to compare MnM to alternative model learning approaches.
> * Figure 10: We have added a new figure showing how the MSE of the MnM model evolves throughout training. This analysis shows that the training dynamics of MnM are relatively stable, even though the method resembles a GAN. While the MnM model produces better policies than the maximum likelihood model, this plot shows that the MnM model actually has worse accuracy that the maximum likelihood. This finding suggests that our joint objective enables training to focus on overall RL performance, rather than just model accuracy.
>
> > The significance of the paper is limited...  Even a standard MLE-based MBRL algorithm optimizes a lower bound on the expected returns.
>
> Thank you for pointing this out. We have added a paragraph to the revised paper (paragraph 3 in Section 2) to note that the Simulation Lemma (and other prior work) also optimize a lower bound on expected returns. Our lower bound is arguably more useful than the bound from the Simulation Lemma for two reasons. First, our bound can be readily *estimated*. It is unclear how to practically optimize the bound from the Simulation Lemma in MDPs with continuous states and actions; that would require guaranteeing that the model error on every state and action be less than some threshold. In contrast, our lower bound involves an expectation, which can be estimated using samples. A second reason our bound may be more useful is that our bound can be readily *optimized*. Whereas our method jointly optimizes the model and the policy using the same objective, the method proposed in the Simulation Lemma paper involves updating the model and the policy using different objective; indeed, it is unclear how to directly optimize the bound from the Simulation Lemma in MDPs with continuous states and actions.
>
> > random seeds
>
> All plots in the paper were run for at least 3 random seeds; we have added this detail to Appendix C. To address the reviewer's concern about reproducibility, we reran experiments on one of the environments from Figure 5 ('DClawScrewFixed-v0'), this time using 5 random seeds. The results, shown in Figure 8, result in the same conclusions as before.
>
> As suggested by the reviewer, we will attempt to run more seeds on more tasks. However, as each experiment takes approximately 72 hours to complete, running 10 seeds on 12 tasks with 3 baselines would involve about 26,000 hours of compute.
>
> > In tabular experiments, there is no comparison to an MLE MBRL agent.
>
> In the tabular experiments (Figure 2a), a MLE MBRL agent that learned a perfect model would be identical to the "Q learning" baseline. We have revised the plot to call this method "Correct Model." MnM outperforms even this oracle version of the MLE MBRL agent.
>
> > In continuous control, there is no comparison to any MBRL algorithm that addresses the objective mismatch.
>
> As suggested by the reviewer, we have added comparisons to three additional MBRL algorithms that address objective mismatch. On the benchmark locomotion tasks, we compare to VMBPO (red curves in Figures 5), finding that it performs worse than MnM. We choose these tasks for the comparison with VMBPO because VMBPO reports numbers on these tasks.
>
> We then compare to VAML [3] and the value-weighted maximum likelihood loss proposed in [2]. We use the more difficult DClawScrewFixed-v0 task for the comparison to the other methods, as the higher dimensionality (36 dimensional observations) and complex contact dynamics of this task are more likely to cause model exploitation. We report results in new Figure 8, which uses 5 random seeds. MnM outperforms these alternative model learning approaches.

---

> > ### Author Response · Authors · 2021-11-18
> > **Response to D2vZ**
> >
> > **[response 2 of 3]**
> >
> > > Lastly, while MnM mitigates the objective mismatch, it is unclear from the results what are the benefits of this.
> >
> > The main contribution of this paper is an objective for model based RL with appealing theoretical guarantees. Our four didactic experiments probe and experimentally verify these theoretical guarantees. These experiments show that the benefits of MnM include faster learning (Figure 2a), better handling of inaccurate models (Figure 2b), and optimizing a lower bound (Figure 3b). Because these experiments use simple environments, we can directly study whether the proposed *objective* is useful, independent of the particular *optimization* procedure. The proposed objective can be optimized using any planning or RL algorithm.
> >
> > To try to directly address the reviewer's concerns about the number of random seeds in Figure 3, we repeated the experiment on one of the environments where the reviewer had concerns about statistical significance (DClawScrewFixed-v0) using 5 random seeds. The results, shown in Figure 8, are nearly identical to the original results shown in Figure 3, supporting the reproducibility of our results.
> >
> > > The new bound relies on (a) non-markovian dynamics (b) time-varying and parameterized discount factor, limiting the justification of the method by this theoretical result.
> >
> > Our practical implementation of MnM optimizes the simpler lower bound from Eq. 2 (first sentence of Section 4). We believe that optimizing a lower bound is a strong justification for a model-based RL algorithm (most model-based RL algorithms do not do this). We do not claim that our practical implementation optimizes the more complicated lower bound from Eq. 4. To clarify this point, we have revised part of the third paragraph of the introduction and the final paragraph of "An objective for model-based RL."
> >
> > While we have not implemented an algorithm to optimize the more complex lower bound, prior work has successfully learned non-Markovian dynamics [6] and (separately) learned discount factors [7]. Thus, optimizing the more complex lower bound may be feasible using architectures and tricks from prior work.
> >
> > > Since the bound that the practical algorithm optimizes is not tight, it is unclear to which extent the proposed method mitigates the objective mismatch.
> >
> > There might be a confusion in how we are using the term "objective mismatch." We use this term to mean that the model and policy are trained with the same objective, regardless of how that objective relates to the true expected return objective. This property is useful because it means that gradient updates to the model will also make the policy better, according to this objective. In contrast, gradient updates to a maximum likelihood model can make the policy worse. MnM mitigates the objective mismatch problem because the model and policy are trained using the same objective.
> >
> > Importantly, there is a (provable) relationship between the proposed objective and the expected return: the proposed objective is a lower bound on the expected return. This is practically useful because it guarantees users that a policy trained with the model will do well when deployed in the true environment. Prior work that proposes lower bounds that are either very difficult to compute [8] or only hold for a small region of "similar" models and policies [1].
> >
> > > Lastly, the analyses make several assumptions that might not be entirely justified. The most noticeable one is the assumption about the positive reward. While it is common to assume that the rewards are non-negative, the assumption about positive rewards might be limiting.
> >
> > The assumption that reward functions are bounded is standard in the literature [4, 5]. Of course, we can add a constant to make any bounded reward function positive, which does not change the optimal policy.
> >
> > > Moreover, the rewards in certain environments (e.g. HalfCheetah-v2) where MnM is tested take negative values
> >
> > As noted in Section 5.2, we did not apply the logarithmic transformation to the robotic control experiments. We have revised this sentence to emphasize this point. Of course, computing the logarithm of a negative reward would not make sense.
> >
> > > The literature review could be improved.
> >
> > We have added the suggested references to the related work section.

---

> > > ### Author Response · Authors · 2021-11-18
> > > **Response to D2vZ**
> > >
> > > **[response 3 of 3]**
> > >
> > >
> > > > Since $\Gamma_\theta(t)$ is a CDF, does it imply that $\gamma(t)$ is a distribution?
> > >
> > > Yes, $\gamma_\theta(t)$ is a distribution. We have added a sentence to clarify this in Section 3.
> > >
> > > > “One detail of note is that we omit the reward augmentation for MnM during these experiments, as it hinders exploration leading to lower returns.” Is it the entropy augmentation of SAC or the proposed augmentation with the log density ratio?
> > >
> > > This is the proposed augmentation with the log density ratio (Equation 3). We have clarified this sentence to refer to the transformation in Equation 3.
> > >
> > > > Is the reward function learned as well?
> > >
> > > Yes, the reward function is learned as well. We fit a separate neural network to predict the rewards (final paragraph of Section 4.2).
> > >
> > >
> > > [1] Luo, Yuping, et al. "Algorithmic framework for model-based deep reinforcement learning with theoretical guarantees." arXiv preprint arXiv:1807.03858 (2018).
> > >
> > > [2] Lambert, Nathan, et al. "Objective mismatch in model-based reinforcement learning." arXiv preprint arXiv:2002.04523 (2020).
> > >
> > > [3] Farahmand, Amir-massoud, Andre Barreto, and Daniel Nikovski. "Value-aware loss function for model-based reinforcement learning." Artificial Intelligence and Statistics. PMLR, 2017.
> > >
> > > [4] Sun, Wen, et al. "Model-based rl in contextual decision processes: Pac bounds and exponential improvements over model-free approaches." Conference on learning theory. PMLR, 2019.
> > >
> > > [5] Shah, Devavrat, et al. "Sample efficient reinforcement learning via low-rank matrix estimation." arXiv preprint arXiv:2006.06135 (2020).
> > >
> > > [6] Hafner, Danijar, et al. "Dream to control: Learning behaviors by latent imagination." arXiv preprint arXiv:1912.01603 (2019).
> > >
> > > [7] Rudner, Tim GJ, et al. "Outcome-Driven Reinforcement Learning via Variational Inference." arXiv preprint arXiv:2104.10190 (2021).
> > >
> > > [8] Kearns, Michael, and Satinder Singh. "Near-optimal reinforcement learning in polynomial time." Machine learning 49.2 (2002): 209-232.

---

> > > > ### Author Response · Authors · 2021-11-22
> > > > **Response to D2vZ**
> > > >
> > > > Dear Reviewer,
> > > >
> > > > We hope that you've had a chance to read our response. We would really appreciate a reply as to whether our response and clarifications have addressed the issues raised in your review, or whether there is anything else we can address.

---

> > > > > ### Comment · Reviewer_D2vZ · 2021-11-23
> > > > > **Further discussion**
> > > > >
> > > > > Dear Authors,
> > > > >
> > > > > Thank you for the reply and detailed explanations.
> > > > >
> > > > > **The revisions do not properly address the major concern about the reliability and the reproducibility of the results.**
> > > > >
> > > > > I appreciate the new results in Figure 8 studying the comparison of MnM with 4 baselines on `DClawScrewFixed-v0` domain using 5 random seeds.
> > > > >
> > > > > However, these results are given **only for 1 single environment while Figure 5 contains 11 other domains where results are still inconclusive** both because of 3 seeds (combined with high variance of returns) and lack of benchmarks (VMBPO for CartPole, MetaWorld, and DClaw beyond ScrewFixed-v0).
> > > > >
> > > > > I appreciate the computational complexity of running the experiments using more than 3 seeds. However, given the high variance of results, it is not possible to make conclusions whether MnM outperforms the MBPO and SAC baselines.
> > > > >
> > > > > Moreover, while I appreciate the similarity of the bound used in practice and for theoretical analysis, there are significant differences between them (e.g. discount factor being a parameterized distribution instead of a scalar, non-markovian dynamics) limiting the theoretical contribution of the paper.

---

> > > > > > ### Author Response · Authors · 2021-11-23
> > > > > > **p-values and the significance of the simple lower bound**
> > > > > >
> > > > > > Thank you for clarifying the concerns about the paper. This discussion is useful for helping us make the paper more rigorous.
> > > > > >
> > > > > >
> > > > > > > However, given the high variance of results, it is not possible to make conclusions whether MnM outperforms the MBPO and SAC baselines.
> > > > > >
> > > > > > To make conclusions about whether MnM outperforms the baselines, we have gone back and computed p-values associated with each environment. Specifically, we looked at the final reward for each method (i.e., the right side of each plot). This allows us to make the following claims:
> > > > > > * MnM outperforms VMBPO on the three gym locomotion tasks (all p values < 0.04)
> > > > > > * MnM outperforms MBPO on the four DClaw tasks (all p values < 0.01)
> > > > > >
> > > > > > We will revise the paper also note the following negative results:
> > > > > > * MnM does not outperform SAC on all the metaworld and DLaw tasks. The p-values are 0.3 for metaworld-drawer-v2, 0.33 for DLawTurnRandom-v0, and < 0.08 for all other tasks.
> > > > > >
> > > > > > > The difference between the two lower bounds limits the theoretical contributions of the paper
> > > > > >
> > > > > > Even the simple lower bound, which corresponds to the actual method used in the experiments, is stronger than prior work (see paragraph 3 of Section 2; compare to Luo 2019, Kearns 2002).
> > > > > >
> > > > > > We believe that describing the more complex version of the bound in the text is a net positive improvement to the paper; we could have  omitted the more complex bound and provided only the description of the simple lower bound used in our experiments. We further revised Section 3 to more clearly emphasize that our experiments will optimize the simple bound, and that this paper does not propose a method to optimize the complex bound.

---

### Official Review · Reviewer_7QkN · 2021-11-07

**Correctness:** 3
**Technical Novelty And Significance:** 3
**Empirical Novelty And Significance:** 3
**Recommendation:** 6
**Confidence:** 4

**Main Review:**

Writing: The paper is written clearly, although it has a few typos.

1. The link to the proof of Lemma 3.2 is wrong.
2. Eq (8): What is $\tilde Q_\phi$? Is it $Q_\psi$​?
3. Appendix A.3, (d), the first cancelled term should be $\log \pi_\theta(a_t | s_t)$​.
4. Appendix A.1, "Rather, it corresponds to maximizing a sum of the expected return **and** the *variance* of the return"

Novelty: The proposed method is similar to a prior work,  VMBPO. Both share similar components, with similar method of learning. However, there are a few difference which separates them apart. One of the most important differences is that MnM takes the logarithm of rewards, and derives a lower bound of the log of expected return, while VMBPO derives a variational lower bound of an upper bound of the log of the expected return. Given VMBPO as a prior work, The theoretical result is not so surprising. If the technique of logarithm of reward can be well justified, I think the empirical novelty is good.

Other comments:

> Following prior work (Janner et al., 2019), we learn an additional model to predict the true environment.
What does it mean by "additional model"?

Is the dynamics model learned well? From my personel research experience, learning the model using a GAN-like objective leads to an inaccurate model and sometimes produces unrealistic states. The signals from the discriminator are much weaker than the signals from supervised learning. What if we train $C$​ and $\pi$​ with MnM, but train maximum likelihood model $q$​​?

In Figure 6: What does it mean by "For fair comparison, we use Q values corresponding to just the task reward"? MBPO uses $r$ while MnM uses $\log r$, so they indeed differ a lot.

As the new reward function $\tilde r$ takes the logarithm of old reward function $r$, it seems that (a) it's tricky to apply logarithm to non-positive numbers; (b) The new reward function $\hat r$  is not invariant to the scale of the old reward function $r$, i.e., shifting $r$ can lead to a very different $\tilde r$. Do all these $\tilde r$​ work in practice?

Theoretically, there is indeed a difference whether we transform the reward. But I'm not very convinced that this technique is essential in practice. The authors constructs a 3-state MDP in Figure 3, but some details are missing, e.g., how is $\eta$​​​​ in VMBPO chosen? I agree that VMBPO optimizes an upper bound approximately, but it can somehow control overestimation by adjusting $\eta$​, while I don't see how MnM controls underestimation. Moreover, what happens if we don't apply this technique in practice? I expected to see the ablation in a more complex environment.



**Summary Of The Paper:**

This paper studies the the problem of objective mismatch problem in model-based reinforcement learning, which states that the goal of the model is different from the goal of policy optimization. To solve this issue, this paper proposes a single objective to train both the policy and the dynamics model, by deriving a lower bound of the (log of) expected return and optimizing the lower bound jointly. Based on the objective, the paper proposes the Mismatched no More (MnM) algorithm, which use a GAN-style objective to train the model. Experiments show that the proposed algorithm can match the performance of baseline algorithm (MBPO) when exploration is not a big issue, while outperforms it significantly for sparse-reward tasks.

**Summary Of The Review:**

This paper is written well and the idea sounds very interesting and promising. However, as stated in the review, I'm not fully concerned by the log reward technique. Overall, I think there is more merit in this paper than the flaw so I recommend weak acceptance.

---

> ### Author Response · Authors · 2021-11-18
> **Response to 7QkN**
>
> **[response 1 of 2]**
> We thank the reviewer for their detailed review. The reviewer's main concerns seem to be a number of minor experimental and writing questions. We have addressed or answered all the questions below. **Do these revisions to the paper, together with the discussion below, fully address the reviewer's concerns of the paper?**
>
> > Given VMBPO as a prior work, the theoretical result is not so surprising.
>
> We believe that the theoretical results of our paper are substantially different from VMBPO. While both papers propose solutions to the objective mismatch problem, only ours maximizes a lower bound on the expected return.
>
> Maximizing a lower bound (as done by our method) is usually preferable to maximizing an upper bound. A lower bound gives users confidence that an agent's performance in the real world will be at least as good as the performance under the learned model. In contrast, an upper bound tells the user that the agent will perform worse in the real world than under the learned model, without indicating how much worse it might perform. Maximizing an upper bound could lead to the true return not changing at all, but its (erroneous) estimate increasing arbitrarily. VMBPO does not provide any theoretical guarantees on how much overestimation may occur (for finite $\eta$).
>
> The theoretical results of our paper should also be evaluated in light of prior work that propose lower bounds for model-based RL. We have added a new paragraph to highlight this comparison (paragraph 3 in Section 2). To the best of our knowledge, our paper is the first to propose an objective for model based RL that has two important theoretical properties
> 1. The objective provably a global lower bound on expected return that can be computed in MDPs with continuous states and actions
> 2. The objective jointly optimizes the model and policy using the same objective.
>
> Perhaps the most similar prior work [1] also proposes an objective that is a lower bound, but it doesn't satisfy either of these properties: the bound in [1] only holds locally, and it involves optimizing the model and policy using different objectives. This prior work even suggests that deriving a global lower bound (as done by our method) may be "impossible."
>
> > Is the dynamics model learned well?
>
> The MnM dynamics model does train stably and its MSE decreases throughout training. As expected, the MSE of the MnM dynamics model is larger than the MSE of a maximum likelihood model. So, while the MnM dynamics model may not be as accurate as the maximum likelihood model, it may be more useful, as it sometimes results in higher return policies. We have revised the paper to add a figure (Figure 10) showing the MSE of the MnM dynamics model throughout training.
>
> > I don't see how MnM controls underestimation
>
> You are right that the MnM objective does not have a single parameter to control underestimation. We have added a sentence to Section 6 to emphasize this limitation.
>
> The more complex lower bound (Eq. 4) can become tight at optimality, meaning that the underestimation can be driven to zero if we have sufficiently expressive function approximators. The simpler lower bound (Eq. 2) is not necessarily tight at optimality. While we currently don't have a bound on how large this underestimation can be, our empirical experiments suggest that optimizing even this simpler lower bound results in optimal or near-optimal policies.
>
> One way to measure the degree of underestimation would be to compare our lower bound to an upper bound on expected return, such as the objective proposed in VMBPO. If this gap is small, then both objectives are close to the true expected return. However, if the gap is large, it is unclear whether one or both of the objectives are far from the true expected return.

---

> > ### Author Response · Authors · 2021-11-18
> > **Response to 7QkN**
> >
> > **[response 2 or 2]**
> > ### Reward Transformation
> >
> > > As the new reward function $\tilde{r}$ takes the logarithm of old reward function $r$, it seems that (a) it’s tricky to apply logarithm to non-positive numbers; (b) The new reward function $\hat{r}$ is not invariant to the scale of the old reward function $r$, i.e., shifting $r$ can lead to a very different $\tilde{r}$.
> >
> > (a) Yes, we assume that the task rewards are positive (see first paragraph of Section 3). (b) Scaling the original reward function by a constant $c$ results in adding a constant $\log c$ to the modified reward function.
> >
> > > Do all these $\tilde{r}$ work in practice?
> >
> > No, they do not all work in practice. As noted in Section 5.2, our continuous control experiments omit the log transformation because we found it to hurt performance on these tasks. While our objective then becomes similar to VMBPO, we find that in practice our method works quite a bit better (see the new comparison with VMBPO in Fig. 5), likely because of differences in how the model is trained (VMBPO uses a 2-step EM procedure).
> >
> >
> > > Moreover, what happens if we don't apply [the logarithmic transformation] in practice? I expected to see the ablation in a more complex environment.
> >
> > As noted in Section 5.2 (first paragraph), the continuous control experiments already don't apply this logarithmic transformation; we found that it hurt exploration on these domains. This result raises an interesting direction for future work: how can we design model-based RL algorithms for complex domains that explore efficiently while still maximizing a provable lower bound on expected return?
> >
> > ### Writing clarifications
> >
> > > “For fair comparison, we use Q values corresponding to just the task reward, omitting the augmented reward used by MnM.”. MBPO uses $r$ while MnM uses $\log⁡ r$, so they indeed differ a lot.
> >
> > For this plot, we show the Q values corresponding to the task reward, without any logarithmic transformation. We have revised the text to clarify this.
> >
> > > [VMBPO] can somehow control overestimation by adjusting $\eta$
> >
> > We note that VMBPO maximizes an upper bound for any finite choice of $\eta$.
> >
> > > Following prior work (Janner et al., 2019), we learn an additional model to predict the true environment. What does it mean by "additional model"?
> >
> > We mean that we train another neural network (i.e., different from the dynamics model and the classifier) that takes as input the state and action and outputs the predicted reward. We have revised this sentence to clarify this point.
> >
> > > how is η​​​​ in VMBPO chosen?
> >
> > We use $\eta = 1$. We have added this detail to Appendix C.1.
> >
> > > The link to the proof of Lemma 3.2 is wrong.
> >
> > > Eq (8): What is $Q_\phi$? Is it $Q_\psi$​?
> >
> > > Appendix A.3, (d), the first cancelled term should be $\log \pi_\theta(a_t \mid s_t$​.
> >
> > > Appendix A.1, "Rather, it corresponds to maximizing a sum of the expected return and the variance of the return"
> >
> > We have fixed these typos. Thanks for bringing them to our attention.
> >
> > [1] Luo, Yuping, et al. "Algorithmic framework for model-based deep reinforcement learning with theoretical guarantees." arXiv preprint arXiv:1807.03858 (2018).

---

> > > ### Author Response · Authors · 2021-11-22
> > > **Response to 7QkN**
> > >
> > > Dear Reviewer,
> > >
> > > We hope that you've had a chance to read our response. We would really appreciate a reply as to whether our response and clarifications have addressed the issues raised in your review, or whether there is anything else we can address.

---

> > > ### Author Response · Authors · 2021-11-24
> > > **Any additional questions/concerns?**
> > >
> > > Dear Reviewer,
> > >
> > > Thank you for raising a number of questions and concerns in the initial review. Revising the paper to address these concerns has made the paper more precise and rigorous. Have the revisions and responses above addressed all the reviewer's concerns? We would be happy to address any additional questions.

---

### Decision · Program_Chairs · 2022-01-20

**Decision:**

Reject

**Comment:**

The reviewers agree that the proposed method of joint model-policy optimization using a lower bound is novel and interesting and worthwhile pursuing. But all reviewers find a variety of issues in the paper, such that ratings are just above borderline or below. Given all the mixed feedback, it appears that the paper is still a bit premature for publication and could greatly benefit from improvements in a future submission.